# Electron penetration triggering interface activity of Pt-graphene for CO oxidation at room temperature

Yong Wang[1,2,7], Pengju Ren[3,4,7], Jingting Hu[1,2], Yunchuan Tu[1,2], Zhongmiao Gong[5], Yi Cui[5], Yanping Zheng[1], Mingshu Chen[1], Wujun Zhang[6], Chao Ma[6], Liang Yu[2], Fan Yang[2], Ye Wang[1], Xinhe Bao[2] & Dehui Deng[1,2✉]

Achieving CO oxidation at room temperature is significant for gas purification but still challenging nowadays. Pt promoted by 3*d* transition metals (TMs) is a promising candidate for this reaction, but TMs are prone to be deeply oxidized in an oxygen-rich atmosphere, leading to low activity. Herein we report a unique structure design of graphene-isolated Pt from CoNi nanoparticles (Pt|CoNi) for efficiently catalytic CO oxidation in an oxygen-rich atmosphere. CoNi alloy is protected by ultrathin graphene shell from oxidation and therefore modulates the electronic property of Pt-graphene interface via electron penetration effect. This catalyst can achieve near 100% CO conversion at room temperature, while there are limited conversions over Pt/C and Pt/CoNiO$_x$ catalysts. Experiments and theoretical calculations indicate that CO will saturate Pt sites, but O$_2$ can adsorb at the Pt-graphene interface without competing with CO, which facilitate the O$_2$ activation and the subsequent surface reaction. This graphene-isolated system is distinct from the classical metal-metal oxide interface for catalysis, and it provides a new thought for the design of heterogeneous catalysts.

[1] State Key Laboratory of Physical Chemistry of Solid Surfaces, Collaborative Innovation Center of Chemistry for Energy Materials (iChEM), College of Chemistry and Chemical Engineering, Xiamen University, 361005 Xiamen, China. [2] State Key Laboratory of Catalysis, iChEM, Dalian Institute of Chemical Physics, Chinese Academy of Sciences, 116023 Dalian, China. [3] State Key Laboratory of Coal Conversion, Institute of Coal Chemistry, Chinese Academy of Sciences, 030001 Taiyuan, China. [4] National Energy Center for Coal to Liquids, Synfuels China Co. Ltd., 101407 Beijing, China. [5] Vacuum Interconnected Nanotech Workstation, Suzhou Institute of Nano-Tech and Nano-Bionics, Chinese Academy of Sciences, 215123 Suzhou, China. [6] Center for High Resolution Electron Microscopy, College of Materials Science and Engineering, Hunan University, 410082 Changsha, China. [7] These authors contributed equally: Yong Wang, Pengju Ren. ✉email: dhdeng@dicp.ac.cn

Pt-catalyzed CO oxidation to $CO_2$ has a wide range of applications, such as indoor air cleaning, vehicle exhaust treatment, and hydrogen purification in fuel cells. Moreover, as a probe reaction in surface science, it promotes the development of heterogeneous catalysis[1,2]. It is well documented that CO adsorbs more strongly on bare Pt surfaces than $O_2$. Therefore, the activation of $O_2$ on Pt is difficult due to the saturated coverage of CO and is the rate-determining step of the reaction[3,4]. An approach to facilitate this step is to construct an active interface between Pt and 3d TMs oxides[4,5], at which $O_2$ can preferentially adsorb and dissociate into atomic O for reaction. However, such an interface usually requires a hydrogen-rich atmosphere to keep TMs oxides in the coordinatively unsaturated state, such as $FeO_{1-x}$[4,6], $CoO_{1-x}$[7,8], or $NiO_{1-x}$[9,10], otherwise, it will be deactivated by the deep oxidation of TMs in an oxygen-rich atmosphere.

Recently, a general strategy of graphene encapsulating TMs has been developed to protect TMs from oxidation and corrosion[11,12]. It has been found that graphene encapsulating iron-based catalysts exhibit high activity and stability in electrocatalytic oxygen reduction[13,14], oxygen or hydrogen evolution[15–18], and other reactions[19,20] under harsh conditions (strongly acidic/alkaline media and high overpotential). Their excellent catalytic performances are derived from the unique electron penetration effect[12,13], that is, TMs in the metallic state do not need to directly contact the reactants but only employ their electrons to activate the outer graphene for reaction. Inspired by this, herein we design a unique graphene-isolated Pt from CoNi nanoparticles (Pt|CoNi) catalyst, of which CoNi can be protected by graphene from oxidation in an oxygen-rich atmosphere and thus effectively trigger the interface activity of Pt–graphene for CO oxidation at room temperature.

## Results and discussion

**Construction of Pt on graphene encapsulating CoNi alloy.** Graphene encapsulating CoNi alloy was prepared by chemical vapor deposition from $CH_3CN$ or $CH_4$ on CoNi nanoparticles (NPs). The obtained samples are labeled as CoNi@NC and CoNi@C, respectively, according to the carbon sources of $CH_3CN$ and $CH_4$. Then, Pt NPs were synthesized in the form of sol and deposited on them and two control samples, commercial carbon nanotube (CNT) and carbon black (CB), with the same Pt loading amount of 4 wt%. The correspondingly obtained catalysts are labeled as Pt/CoNi@NC, Pt/CoNi@C, Pt/CNT, and Pt/CB, respectively.

The morphology of these catalysts was characterized by a transmission electron microscope (TEM). As shown in Supplementary Fig. 1, CoNi NPs (5–7 nm) are fully encapsulated by single-layer graphene in both CoNi@NC and CoNi@C, and CNT is concomitantly generated in CoNi@NC (Fig. 1a). After loading Pt NPs (1–2 nm) on them, CoNi NPs are still in the metallic state as indicated from their X-ray diffraction (XRD) patterns (Fig. 1e). For Pt/CoNi@NC, it is interesting to find that Pt NPs are selectively deposited on the surface of graphene with CoNi NPs inside but rarely on the surface of the concomitant CNT (Fig. 1b and Supplementary Fig. 2b), indicating that there is strong adsorption of Pt NPs on graphene benefited from the inner CoNi NPs. From Fig. 1c, one can clearly see that Pt and CoNi NPs are isolated from each other by single-layer graphene. The graphene layer cannot be observed from the high-angle annular dark-field scanning TEM (HAADF-STEM) image (Fig. 1d); thus, a clear gap can be seen between Pt and CoNi NPs. (TEM images of other catalysts see Supplementary Fig. 2.)

**Verification of electron penetration effect.** To verify the electron penetration effect of this composite catalyst, the electronic structure of Pt in each catalyst was detected by synchrotron-based X-ray absorption near edge structure spectroscopy (XANES). Figure 1f is the XANES spectra of Pt $L_{III}$ edge ($2p \rightarrow 5d$ transition) for Pt/CoNi@NC, Pt/CoNi@C, Pt/CNT, and Pt/CB (original data see Supplementary Fig. 3). Pt on CoNi@NC and CoNi@C display a lower edge jump than Pt on CNT and CB, indicating a higher d-band electron density for the former[21]. The d-hole count of Pt in each catalyst (Fig. 1g) was calculated from the integral area beneath the white line region within $E_0 \pm 10$ eV in the XANES spectrum by assuming the d-hole count of Pt foil and $PtO_2$ to be 1.334 and 4, respectively[21]. The d-hole count of Pt on graphene encapsulating CoNi alloy is 0.33 lower than that on pure carbon materials (CNT and CB), indicating that electrons of CoNi NPs can penetrate through graphene to Pt NPs.

Two comparable catalyst models were established to further confirm the electron penetration effect (Fig. 1h). The $Pt_4$ on the hollow graphene cage exhibits a positive Bader charge of +0.15 due to the electron transfer from Pt to C, whereas the $Pt_4$ on the graphene cage with CoNi NPs inside exhibits a negative charge of −0.08, indicating that CoNi does indeed transfer electrons to Pt through graphene. Benefiting from the electron penetration effect, the binding energy between $Pt_4$ and graphene is strengthened from −2.17 to −2.74 eV when encapsulating CoNi NPs into the graphene cage. This can explain the selective deposition of Pt NPs on the surface of graphene encapsulating CoNi alloy shown in Fig. 1b and Supplementary Fig. 2b, which is due to the enhanced interaction between graphene and Pt by the electron penetration effect. The same conclusion can also be drawn from other models with larger Pt clusters (Supplementary Fig. 4 and Supplementary Table 1).

The electron penetration effect can affect not only the electronic structure of Pt but also that of C atoms in graphene. Figure 1i shows the projected density of state (PDOS) of $2s + 2p$ orbitals of C in $Pt_4$/CoNi@C and $Pt_4$/C (Supplementary Fig. 5 for other models). After introducing CoNi NPs into the graphene cage, the Fermi level of C raises from −4.4 to −3.8 eV, the PDOS near the Fermi level increases, and the PDOS below the Fermi level becomes smooth. These changes mean that the local work function of graphene is decreased[13], which can also be confirmed by the result of the near ambient pressure X-ray photoelectron spectroscopy (NAP-XPS)[22] shown in Supplementary Fig. 6. In other words, graphene turns to be more active when encapsulating CoNi NPs. Furthermore, by comparing the differential charge density between $Pt_4$/CoNi@C and $Pt_4$/C (Fig. 1j), one can see that the charge rearrangement of $Pt_4$/CoNi@C is more significant, especially around the Pt–graphene interface. It illustrates the contribution of the encapsulated CoNi NPs towards making the charge distribution at the Pt–graphene interface more polarized.

**Enhanced performance of CO oxidation by electron penetration.** Temperature-programmed CO oxidation was performed over the four catalysts (Fig. 2a). Compared with Pt/CNT and Pt/CB, Pt/CoNi@NC and Pt/CoNi@C exhibit much higher activity. Near 100% CO conversion can be reached at room temperature. The long-term stability of Pt/CoNi@NC was also tested at 25 °C (Supplementary Fig. 7). The CO conversion only declines by 12% after 24 h, which is superior to those over $Pt–FeO_{1-x}$/CB, $Pt–CoO_{1-x}$/CB, and $Pt–NiO_{1-x}$/CB catalysts (40–60% drop after 24 h at room temperature)[7], and the activity can be fully recovered after $H_2$ reduction (Supplementary Fig. 7) due to the fact that the catalyst structure did not change after the 24-h run (Supplementary Fig. 8). Therefore, the strategy of graphene encapsulating TMs is effective to improve the activity and durability of Pt-catalyzed CO oxidation reaction in an oxygen-rich atmosphere.

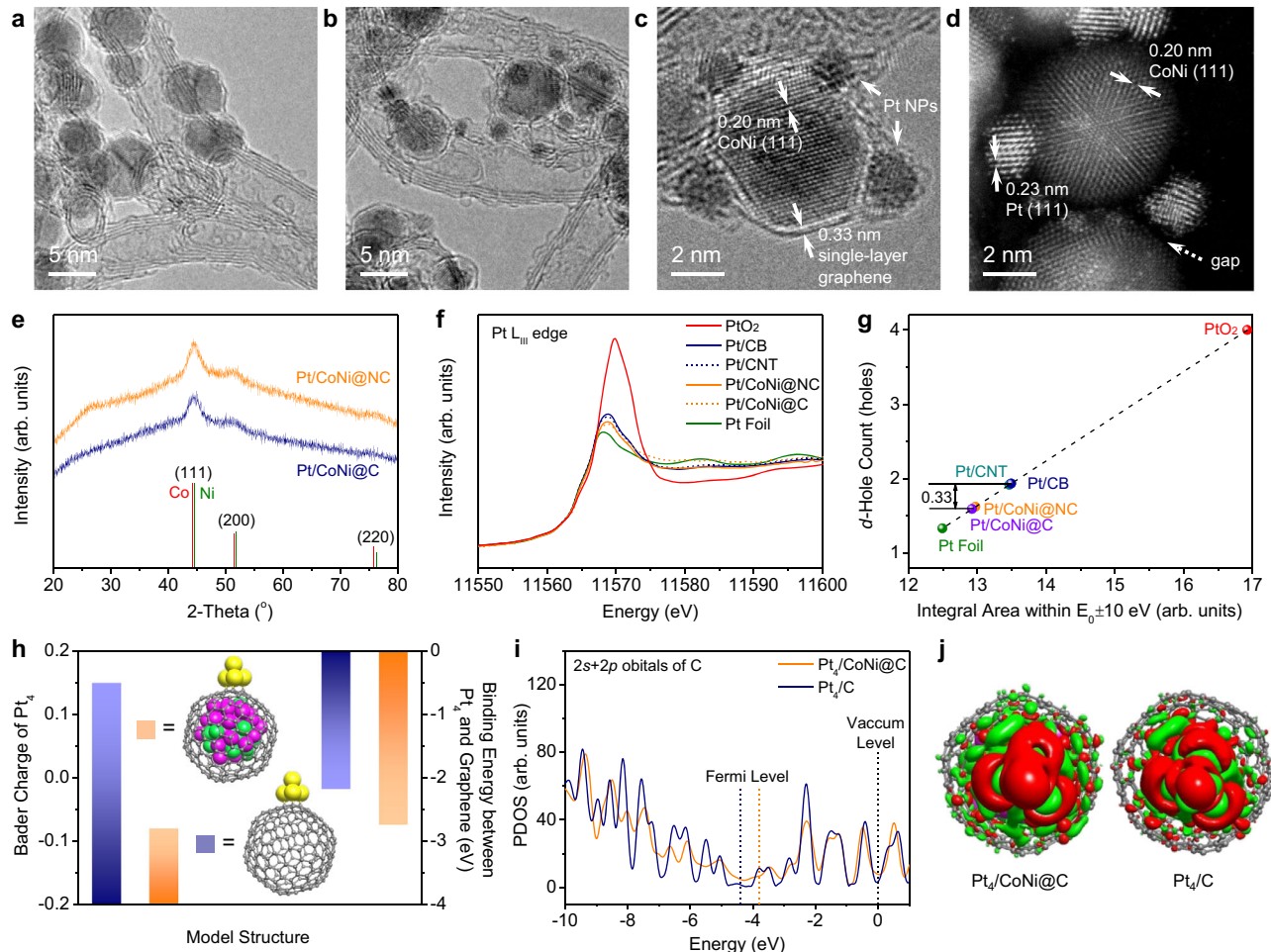

**Fig. 1 Structural characterization and modeling of Pt on graphene encapsulating CoNi alloy. a** TEM image of CoNi@NC. **b, c** TEM images of Pt/CoNi@NC. **d** HAADF-STEM image of Pt/CoNi@NC. **e** XRD patterns of Pt/CoNi@NC and Pt/CoNi@C. **f** XANES spectra of Pt $L_{III}$ edge for Pt/CoNi@NC, Pt/CoNi@C, Pt/CNT, and Pt/CB. **g** $d$-hole count of Pt in each catalyst calculated from their XANES spectra. **h** Catalyst models of $Pt_4$ on graphene cages with and without CoNi NPs inside and the corresponding Barder charge of $Pt_4$ (columns on the left) and the binding energy between $Pt_4$ and graphene (columns on the right). **i** PDOS of $2s + 2p$ orbitals of C atoms in $Pt_4$/CoNi@C and $Pt_4$/C. **j** Top view of differential charge density of $Pt_4$/CoNi@C and $Pt_4$/C. The red and green regions indicate electron increment and depletion, respectively.

**The important role of intact graphene**. To check the role of graphene in protecting CoNi NPs from oxidation, NAP-XPS was applied to monitor the valence states of Co and Ni in Pt/CoNi@NC during an in situ reaction in a flow of 0.067 mbar CO and 1.13 mbar $O_2$. As shown in Fig. 2b, below 150 °C, the peaks of Co $2p_{3/2}$ and Ni $2p_{3/2}$ are located at 778.5 and 852.8 eV, respectively, corresponding to their metallic states. Above 150 °C, both peaks will shift to higher binding energies, indicating the oxidation of CoNi NPs. It means that the graphene layer will be oxidatively broken above 150 °C, which cannot protect CoNi NPs from oxidation anymore. In addition, there is no obvious change for the signal of Pt $4f_{7/2}$ during the in situ reaction from 25 to 210 °C (Supplementary Fig. 9). Then, an in situ hydrogen reduction was performed to reduce the surface $CoNiO_x$ back to the metallic state (Supplementary Fig. 10), but once exposed to the reaction atmosphere (CO + $O_2$) Co and Ni can be rapidly oxidized even at near room temperature. Therefore, the intact graphene layer is essential to keep CoNi in the metallic state.

To directly observe the break of graphene, TEM characterization on Pt/CoNi@NC samples (Fig. 2c) after temperature-programmed reaction to certain temperatures (120, 160, 200, and 240 °C) was performed. For the sample reacted to 120 °C, its graphene layer is still intact. For the sample reacted to 160 °C, graphene breach can be

observed, and some CoNi NPs leak from the breach and should be oxidized. After reaction at higher temperatures (200 and 240 °C), the graphene layers were mostly or even completely removed, and CoNi NPs were fully oxidized judging from the slightly increased lattice spacing, which is consistent with the NAP-XPS result (Fig. 2b). In a conclusion, the critical temperature for the break of graphene is around 150 °C.

Based on the critical temperature for the break of graphene, three temperature-dependent cycle testings were performed over Pt/CoNi@NC to verify the significance of the metallic CoNi for the high activity toward CO oxidation. As shown in Fig. 2d, the first run was operated from 25 to 120 °C to ensure that CoNi is still in the metallic state after the first run. For the second run, the activity is still as high as the first run. The terminal temperature of the second run was set at 240 °C to decompose the graphene layer and thus oxidize the CoNi NPs. Then a sharp decline of activity can be observed for the third run, and its activity is close to those of intentionally prepared Pt–$CoNiO_x$/C catalysts (Supplementary Fig. 11). From the cycle testings, we can learn that keeping CoNi in the metallic state is essential for the high activity of Pt/CoNi@NC, and the oxidation of CoNi cannot enhance the catalytic activity anymore, since $CoNiO_x$ has no more free electrons to modulate the catalytic properties of the outer graphene and Pt NPs (Supplementary Fig. 12). This experiment

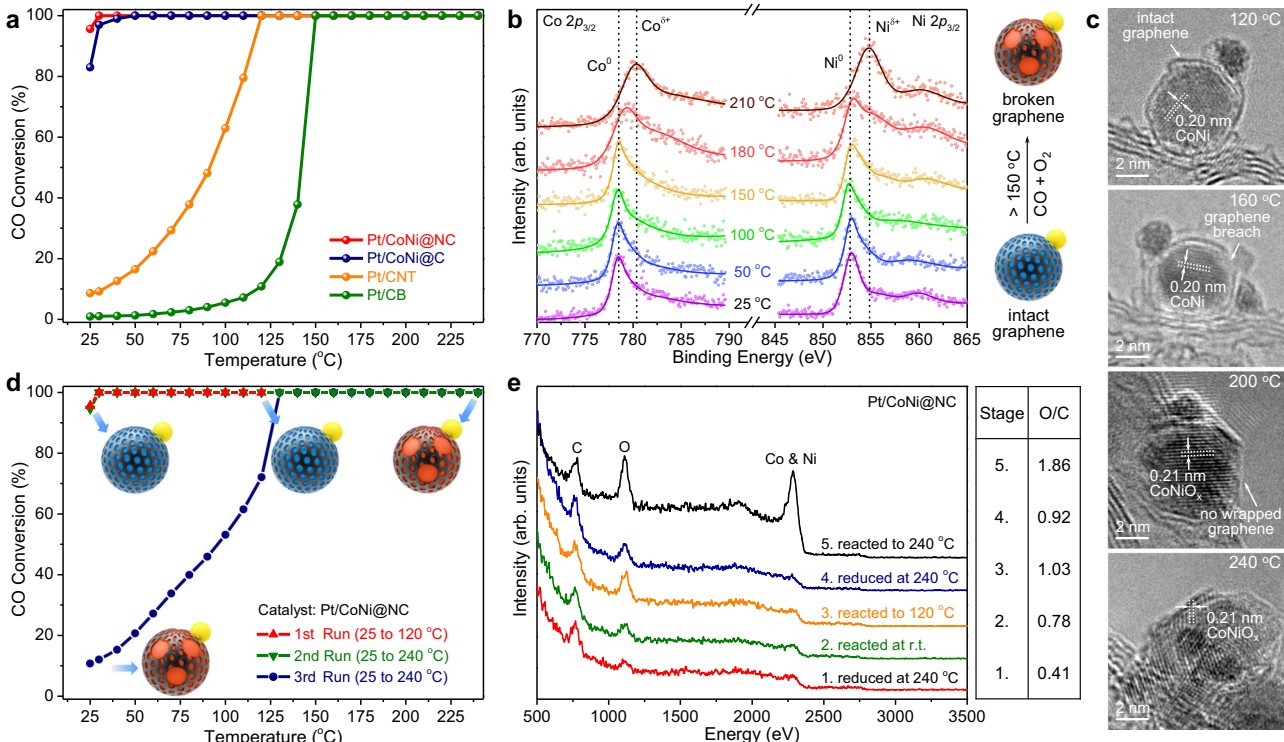

**Fig. 2 Structure–activity relationship of Pt on graphene encapsulating CoNi alloy for CO oxidation. a** Temperature-dependence CO conversion in CO oxidation reaction over the pre-reduced catalysts. 1% CO and 20% O$_2$ in He (1 bar). Space velocity: 60,000 mL g$^{-1}$ h$^{-1}$. **b** XPS spectra of Co 2p$_{3/2}$ and Ni 2p$_{3/2}$ from NAP-XPS testings over Pt/CoNi@NC in a flow of 0.067 mbar CO and 1.13 mbar O$_2$ at different reaction temperatures. **c** TEM images of Pt/CoNi@NC after reacting to certain temperatures. **d** Temperature-dependence CO conversion in CO oxidation cycle testings over Pt/CoNi@NC. 1% CO and 20% O$_2$ in He (1 bar). Space velocity: 60000 mL g$^{-1}$ h$^{-1}$. **e** LEIS spectra of Pt/CoNi@NC after five successive in situ reaction processes. 1% CO and 20% O$_2$ in He (1 bar). Space velocity: 60,000 mL g$^{-1}$ h$^{-1}$. The right table shows the peak area ratio of O to C in catalyst at these stages.

validates the importance of the intact graphene layer for the keep of CoNi in the metallic state and thus for the effectiveness of the electron penetration effect on the room-temperature activity of CO oxidation.

**Surface catalytic reaction mechanism.** To monitor the change in the surface element composition of Pt/CoNi@NC during the reaction, a high-sensitivity low-energy ion scattering spectroscopy (LEIS) was applied. The detection depth of LEIS is only two atomic layers[23]. An in situ reaction chamber is connected with the chamber of LEIS, which allows the reduction and reaction of catalysts with the same parameters of temperature and pressure as for the actual testings. Five successive processes (Fig. 2e) were performed over Pt/CoNi@NC in the in situ reaction chamber. First, the catalyst was reduced at 240 °C for 2 h (Stage 1) and then reacted at room temperature for 2 h (Stage 2). Next, a temperature-programmed reaction was performed from 30 to 120 °C at a rate of 1 °C/min (Stage 3). After that, the catalyst was re-reduced at 240 °C for 2 h (Stage 4). The last process is the temperature-programmed reaction from 30 to 240 °C at a rate of 1 °C/min (Stage 5). LEIS spectra were obtained after each stage. From Fig. 2e, one can see that the CoNi signal (2285 eV) turns from terraces into a sharp peak after reacting to 240 °C (stage 5), indicating the exposure of metals. Therefore, before the break of graphene (stage 1–4), the catalyst surface is mainly constituted of C and O elements, and the detected O should mainly bond with C considering that the surface content of Pt (2750 eV) is insufficient. The peak area ratio of O to C is summarized in the right table of Fig. 2e. Compared with the fresh sample (stage 1), the surface O species gradually increased after reacting at room temperature (stage 2) and to 120 °C (stage 3),

and cannot be effectively removed after re-reducing at 240 °C (stage 4). It indicates that O$_2$ can be activated on the graphene surface even at room temperature, which can also be found over CoNi@NC without loading Pt NPs (Supplementary Fig. 13). Considering that CO needs to be activated on Pt NPs (Supplementary Fig. 14), only the activated O on graphene adjacent to the periphery of Pt NPs can participate in the CO oxidation reaction. Therefore, we suppose that the Pt–graphene interface should be the main active site for CO oxidation at near room temperature.

Density functional theory (DFT) calculations were carried out to further reveal the surface reaction mechanism of CO oxidation over Pt on graphene encapsulating CoNi alloy and the promotion mechanism of the electron penetration effect. Figure 3a shows the adsorption energy of CO and O$_2$ on Pt$_4$/CoNi@C and Pt$_4$/C. For both models, the adsorption energy of CO on Pt site even with a saturated coverage (Supplementary Fig. 15) is much higher than that of O$_2$ (adsorption models see Supplementary Fig. 16), indicating that O$_2$ cannot compete with CO for adsorption on Pt site. However, we found that O$_2$ can adsorb at the Pt–graphene interface of Pt$_4$/CoNi@C (Fig. 3a) without competing with CO, since CO is unable to adsorb at this interface. DFT calculations also show that O$_2$ cannot adsorb at the Pt–graphene interface of Pt$_4$/C. The same results can also be found on a larger model of Pt nano-strip on graphene layer (Supplementary Fig. 17). It indicates that the Pt–graphene interface is activated by the encapsulated CoNi NPs via the electron penetration effect to achieve the exclusive adsorption of O$_2$.

Figure 3b illustrates the reaction mechanism of CO oxidation over Pt$_4$/CoNi@C. The energy profile starts from model I with 4CO adsorbed on Pt$_4$/CoNi@C. Then, a gaseous

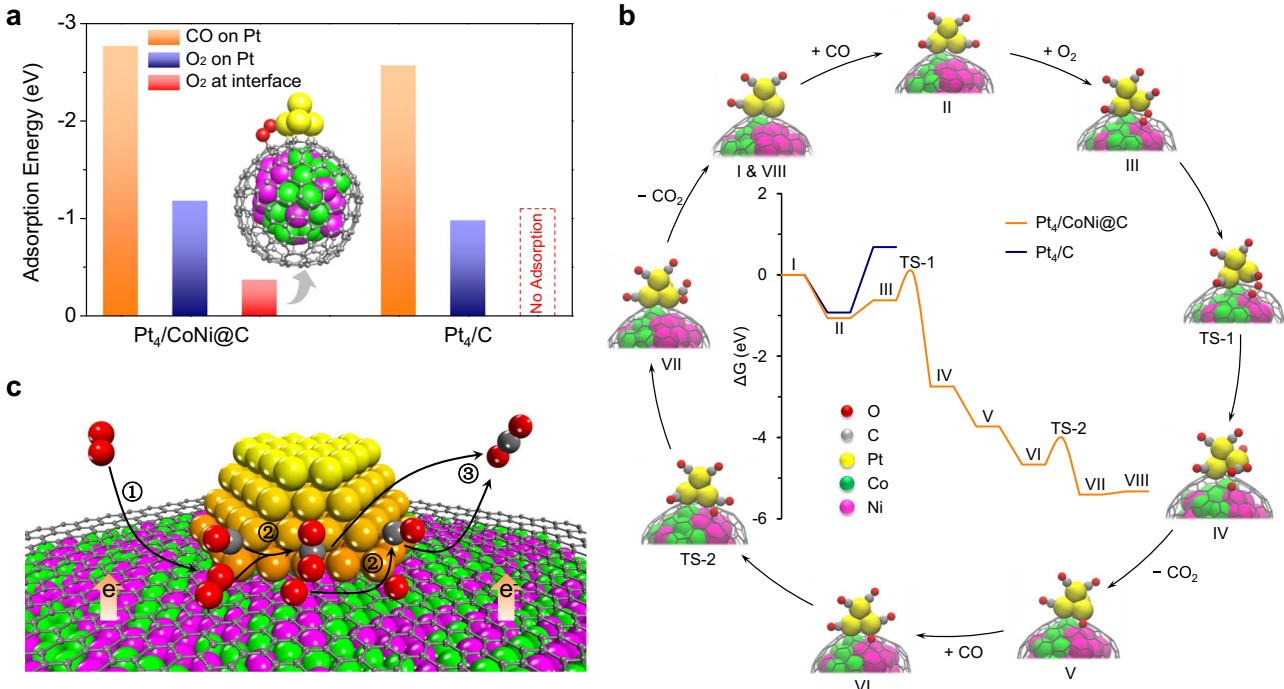

**Fig. 3 DFT calculations on the catalytic mechanism of CO oxidation over Pt on graphene encapsulating CoNi alloy. a** Adsorption energy of CO and $O_2$ on $Pt_4/CoNi@C$ and $Pt_4/C$. **b** CO oxidation energy profiles over $Pt_4/CoNi@C$ (red line) and $Pt_4/C$ (blue line) and the structures of intermediates for $Pt_4/CoNi@C$. "TS" represents transition state. **c** Schematic diagram of electron penetration triggering interface activity of Pt–graphene for CO oxidation at room temperature. The red, grey, yellow, green, and pink balls represent O, C, Pt, Co, and Ni, respectively. (Step 1: $O_2$ adsorption at the Pt–graphene interface, step 2: surface reaction between CO and O, step 3: $CO_2$ release from the Pt–graphene interface).

CO adsorbs on model I preferentially than $O_2$ with a decrease of $\Delta G$. On the 5CO-saturated adsorption structure (model II), $O_2$ can still adsorb at the Pt–graphene interface (model III) with a slight increase of $\Delta G$ (0.44 eV) due to the loss of entropy. By contrast, $O_2$ cannot adsorb at the interface of $Pt_4/$ C but would competitively adsorb on the CO-saturated Pt site with a huge increase of $\Delta G$ (1.61 eV). During the transition state (model TS-1), the activation of $O_2$ is assisted by the adsorbed CO, and then $CO_2$ is formed with a moderate barrier of 0.66 eV. The remaining adsorbed O atom will react with another re-adsorbed CO to form $CO_2$ with a similar barrier of 0.63 eV. Therefore, the graphene encapsulating TMs can avoid the CO poisoning of Pt by providing additional sites for $O_2$ adsorption (Fig. 3c), and the rate-determining step of CO oxidation over our catalyst is no longer the activation of $O_2$ but the surface reaction between CO and O adsorbed at the Pt–graphene interface.

In summary, a highly efficient catalyst of Pt on graphene encapsulating TMs is designed for CO oxidation in an oxygen-rich atmosphere. Such an encapsulation structure usually represents the deactivation of TMs by coke deposit in heterogeneous catalysis. However, in this work, we utilize this structure to protect TMs from oxidation, thereby activating the Pt–graphene interface through the electron penetration effect to realize the exclusive adsorption of $O_2$. Therefore, it changes the rate-determining step from the conventional $O_2$ activation step to a facile surface reaction between CO and O adsorbed at the Pt-graphene interface, which enables a high activity of CO oxidation at room temperature. This graphene–isolated system is a brand new architecture for CO oxidation reaction and beyond. The concept of electron penetration triggering interface activity provides a feasible approach to modulate the performance of catalysts on an electron level.

## Methods

**Preparation of graphene encapsulating CoNi alloy.** First, CoNi layered double hydroxides were deposited on $SiO_2$ sphere (100 nm, Alfa Aesar, Stock No. 43094) by urea co-precipitation with equimolar $Co(NO_3)_2\cdot6H_2O$ (Alfa Aesar) and $Ni(CH_3COO)_2\cdot4H_2O$ (Alfa Aesar). In detail, 1.06 g of $Co(NO_3)_2\cdot6H_2O$, 0.90 g of $Ni(CH_3COO)_2\cdot4H_2O$, and 1.8 g of urea were dissolved into 100 mL of water. After adding 6.0 g of 30% $SiO_2$ sphere colloidal dispersion, the mixture was refluxed at 100 °C for 12 h with continuous stirring. The precipitate was washed with deionized water to neutral and then dried at 100 °C for 12 h. Secondly, the obtained $CoNiO_x/SiO_2$ solid (200 mg) was heated from 30 to 700 °C at a rate of 10 °C/min in a 100 mL/min of 30% $H_2/Ar$ to obtain CoNi metallic alloy. After reaching 700 °C, $H_2$ was switched off and $CH_3CN$ (bubbling with 70 mL/min of Ar) or $CH_4$ (100 mL/min, without Ar) was introduced as carbon sources to form graphene on CoNi NPs. Finally, the cooled sample was immersed into an aqueous solution of 10 wt% HF at room temperature for 12 h and then washed and dried at 40 °C in a vacuum (~0.1 kPa) for 12 h. The obtained samples were labeled as CoNi@NC and CoNi@C corresponding to the carbon sources of $CH_3CN$ and $CH_4$.

**Preparation of Pt/CoNi@NC and Pt/CoNi@C.** First, a colloidal solution of Pt NPs (1–2 nm) was prepared by glycol reduction[24]. In detail, 50 mg of NaOH and 50 mg of $H_2PtCl_6\cdot6H_2O$ were successively dissolved into 5 mL of ethylene glycol, and then it was refluxed at 160 °C for 3 h with continuous stirring under the protection of Ar flow (50 mL/min) to obtain the Pt colloid. Next, the Pt colloid was added into the ethanol dispersion of CoNi@NC or CoNi@C at room temperature. After ultrasonic oscillation for 0.5 h, an aqueous solution of 1 mol/L HCl was added into the mixture to precipitate Pt NPs on CoNi@NC or CoNi@C. Finally, they were thoroughly washed and dried at 40 °C in a vacuum (~0.1 kPa) for 12 h to obtain Pt catalysts for use.

**Preparation of Pt/CNT and Pt/CB.** Commercial CNT (Chengdu Organic Chemicals Co. Ltd.) and CB (Vulcan XC-72) were used for comparison. CNT was pretreated in an aqueous solution of 37 wt% $HNO_3$ at 110 °C for 5 h. CB was directly used without further purification. The same preparation processes were conducted to obtain Pt/CNT and Pt/CB as those for Pt/CoNi@NC and Pt/CoNi@C. The loading amount of Pt for each catalyst was confirmed to be around 4 wt% by inductively coupled plasma–optical emission spectrometry.

**Catalyst characterization.** Transmission electron microscopy (TEM) images were acquired on an FEI Tecnai F20 microscope operating at 200 kV. Aberration-corrected HAADF-STEM images were acquired on a JEOL ARM200F microscope

equipped with a probe-forming aberration corrector operating at 200 kV. XRD measurements were conducted on a Rigaku Ultima IV diffractometer with Cu Kα radiation at 40 kV and 30 mA. X-ray absorption near edge structure spectroscopy (XANES) was measured at the BL14W1 beamline of the Shanghai Synchrotron Radiation Facility. NAP-XPS measurement was carried out by a PHOIBOS 150 NAP 1D-DLD analyzer using monochromatic Al Kα radiation (1486.6 eV). LEIS was obtained using an ion-TOF Qtac100 instrument with helium as an ion source. There is an in situ reaction chamber connecting with the chamber of LEIS[23], which allows the reduction and reaction of catalysts with the same parameters of temperature and pressure as for the actual testings described below.

**Catalyst testing.** CO oxidation was carried out in a quartz tube (6 mm outer diameter and 4 mm inner diameter) reactor. The temperature was measured with a thermocouple on the external surface of the quartz tube right next to the catalyst bed. The reaction gas consisted of 1% CO, 20% $O_2$, and 79% He (Messer). A 20 mL/min of mixed gas flowed through 20 mg of catalyst (200 mesh, <80 μm) to achieve a space velocity of 60,000 mL $g^{-1}$ $h^{-1}$. The tail gas was analyzed online by gas chromatography equipped with a packed column (TDX-01) and a methane converter in front of a flame ionization detector (FID). The methane converter is a micro high-temperature furnace (330 °C) with Ni-based catalyst that can fully convert CO and $CO_2$ into $CH_4$ for FID detection. Before reactions, all catalysts were pre-reduced in high-purity $H_2$ (99.999%, Messer) flow at 240 °C for 2 h. After cooling to room temperature in high-purity Ar (99.999%, Messer) flow, temperature-dependent CO oxidation was performed from 25 to 240 °C (or to 120 °C) at a rate of 1 °C/min.

**DFT calculation.** DFT calculations were performed on the Vienna ab initio simulation package (VASP)[25–27]. The exchange-correlation functional was described as Perdew–Burke–Ernzerhof generalized gradient approximation[28]. The interactions between ionic cores and electrons were described using projector augmented-wave pseudopotentials[29] with kinetics cut-off energy of 400 eV. The vacuum region was set to 15 Å between the clusters. The models of metal clusters encapsulated into graphene ($Co_{27}Ni_{28}$@Gr) consist of $C_{240}$ encapsulating 55 atoms. All structures were fully relaxed to the ground state and spin-polarization was considered in all calculations. The convergences of energy and force were set to $1 \times 10^{-4}$ eV and 0.05 eV/Å, respectively. The free energies of the intermediates were obtained by $\Delta G = \Delta E + \Delta ZPE - T\Delta S$, where $\Delta E$ is the binding energy of adsorption species, and $\Delta ZPE$, $\Delta S$ is the zero-point energy changes, entropy changes, respectively. Totally, 298.15 K was chosen as the temperature according to the experimental condition. The transition states were identified via automated relaxed potential energy surface scans method recently developed by Philipp N. Plessow[30].

## Data availability

All data needed to evaluate the conclusions of this study are available in the main text and Supplementary Information (including the peer review file). The data supporting the findings of this work are available from the corresponding author on reasonable request.

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

## Acknowledgements

We gratefully acknowledge the financial support from the Ministry of Science and Technology of China (Nos. 2016YFA0204100, 2016YFA0200200), the National Natural Science Foundation of China (Nos. 21890753 and 21988101), the Key Research Program of Frontier Sciences of the Chinese Academy of Sciences (No. QYZDB-SSW-JSC020), the Strategic Priority Research Program of the Chinese Academy of Sciences (No. XDB36030200), the China Postdoctoral Science Foundation (No. 2017M622063). We thank the staff at the BL14W1 beamline of Shanghai Synchrotron Radiation Facilities for assistance with the XANES measurements. We thank Prof. Qiang Fu for the fruitful discussion. We thank Ms. Min Hou for her preliminary foreshadowing work.

## Author contributions

D.D. conceived the subject and organized the work. Y.W. and J.H. performed the synthesis and characterization of catalysts and the catalytic performance testings. P.R. and L.Y. contributed to the DFT calculations. Y.T. provided the crucial synthesis method of graphene encapsulating CoNi alloy. Z.G. and Y.C. carried out the NAP-XPS experiments. Y.Z. and M.C. carried out the in situ LEIS and XPS experiments. W.Z. and M.C. conducted the HAADF-STEM characterization. F.Y., Ye W., and X.B. provide valuable suggestions. D.D. modified the paper. Y.W. and P.R. co-wrote the paper and contributed equally to this work.

## Competing interests

The authors declare no competing interests.
