## [Peer Review File · Nature Communications]

REVIEWER COMMENTS

Reviewer #1 (Remarks to the Author):

This manuscript describes unique electron penetration effects from CoNi nanoparticles encapsulated by thin graphene layers, to Pt nanoparticles supported on the surface of the graphene. This complex material is capable of performing CO oxidation at 298 K, whereas carbon-supported Pt particles are not active until higher temperatures. Thorough evidence is included to demonstrate that CoNi are encapsulated (TEM, LEIS), and that the electronic state of Pt is altered by the presence of electron donation from metallic CoNi particles (through the graphene), namely XAS. I thoroughly enjoyed reading the manuscript and feel it presents a wide range of opportunities for future exploration, for those capable of synthesizing similar materials. It was unclear to me if the computational models employed accurately describe the as prepared or working catalyst, both due to the choice of a four-atom Pt cluster to represent 1-2 nm Pt nanoparticles. Additionally, the TEM images shows the possibility of multiple Pt particles adsorbing at a single encapsulated CoNi particle, but the impact of multiple Pt particles on the electron donation from CoNi was not included in the computational models. There are some spectroscopic data that were not included, but may have already been collected by the authors. Some of the experimental methods could be expanded to aid in reproduction of the research. The cause of slow deactivation at 298 K is unclear to me, and additional information could be added on this point. I describe these issues in more detail below. In summary, I applaud the authors on a thorough and intriguing study and recommend it for publication in Nature Communications after minor revisions.

Specific comments:

1. Page 3, Figure 1: Have the authors prepared Pt/(Co, Ni)/C samples, and how does their catalytic performance and spectroscopic characteristics compare to the Pt/CoNi@C materials? I would be interested to know how the d-hole count of Pt from XAS would look on a sample intentionally prepared to have Pt in contact with Co and/or Ni (e.g, perhaps bimetallic PtCo nanoparticles supported on C), as one would expect this might be different from the samples in this paper that have the C layer in between Pt and Co or Pt and Ni. This could provide further evidence that the authors proposed structure is present throughout the bulk of the sample (and not just in the limited regions observed by TEM), and if the spectroscopic data for PtCo particles on C is distinct from those reported in the manuscript, it would provide evidence that Co does not escape from the C shell and alloy with Pt over time. I will note that this is addressed to some extent on page 8, wherein PtCoNiOx is less active than the fresh samples, which the authors propose results from a lack of free electrons in the CoNiOx for donation to Pt.

2. Page 3: The authors mention "it is interesting to find that Pt NPs are selectively deposited on the surface of graphene with CoNi NPs inside but rarely on the surface of the concomitant CNT" and refer the reader to TEM images. The TEM images will only represent a minority of each sample, is there any additional quantitative surface characterization the authors can provide that Pt NPs are selectively deposited near the CoNi NPs? Perhaps IR spectra of adsorbed CO (or Pt XPS) would show a different peak center for Pt/C compared to Pt near encapsulated CoNi? The XANES data show, on average, that Pt/CoNi@C has different edge energy than Pt/C, but these data for Pt/CoNi@C could reflect a mixture of Pt particles distant from CoNi particles and Pt particles adsorbed at CoNi.

3. Page 5 and Figure 1: is the Pt4 model realistic given that the Pt NPs are 1-2 nm in size and likely comprised of >4 Pt atoms? This is my strongest criticism of the paper in multiple areas – whether or not this model is representative of the samples. Were other models with larger numbers of Pt atoms

considered, and were the results similar to those reported here? The effect of the number of Pt atoms on the calculated parameters should be included in the article or the SI. Additionally, the authors model a single Pt₄ particle on each CoNi cluster, but from TEM (e.g., Figure 1d) it appears plausible that multiple Pt particles are attached to each encapsulated CoNi cluster. What is the effect of inclusion of multiple Pt particles on the calculated parameters? I would imagine either a larger number of atoms per particle, or a larger number of Pt particles per CoNi cluster, would disperse the electronic effects to a greater extent, perhaps minimizing the magnitude of the difference relative to C without CoNi encapsulated. These calculated parameters include Bader charge and Pt binding energy (page 5, Figure 1h), PDOS (Figure 1j), DFT calculations (Figure 3).

4. Page 5 and Figure 1h: is Pt binding and Bader charge at an empty C sphere an appropriate comparison for these characteristics when Pt binds to C-encapsulated CoNi particles? When not binding to an encapsulated CoNi, Pt would realistically be bound to an extended C surface, so perhaps a filled C sphere would be a better comparison for these calculations than a hollow C sphere (assuming there are not hollow C spheres in the C supports used by the authors).

5. Page 6: what leads to the deactivation over 24 h, and why can the material be regenerated in H₂? Have the authors performed XPS (or TEM) on the catalyst after 24 h reaction, and how does this compare to that for the fresh catalysts? TEM after exposure to reaction conditions at higher temperatures is reported in Figure 2c, but I did not see the TEM for the materials after 24 h exposure to reaction conditions at 298 K.

6. Page 6: what was observed for Pt during the NAP-XPS experiments? It would benefit the reader if the Pt XPS data during these treatments were included in Figure 2. Furthermore, I would note that these CO and O₂ pressures are much lower than those used for catalysis, so the effect of extended exposure to 1 kPa CO and 20 kPa O₂ on the oxidation states of Pt, Co, and Ni at 298 K is unclear from the data reported in Figure 2b. Could the authors perform operando Pt, Co, and/or Ni XAS to monitor the oxidation states of these elements at reaction conditions?

7. Page 7, Figure 2: for the caption of figure 2e, it would help if the treatment conditions were included in the caption (gas pressures, treatment durations, flowrates per mass catalyst, etc.).

8. Page 9: Figure 2e shows that there is O on the samples after reaction at 298 K, that cannot be removed even after reduction at 513 K. The authors propose this O reflects activation of O₂ on the graphene surface. Is this signal O₂, or O adatoms? In either case, the fact that these are not removed during after reduction at 240 C suggests that these may be structural or strongly bound O atoms, that may not be participating in catalysis. Are these present on the CoNi@C and C materials in the absence of Pt? It was unclear to me if these O are part of the same reservoir of O that is ultimately incorporated in CO₂, or if these O are spectators/unreactive. In the caption of Figure S5, the authors mention that O₂ can be activated on the surface of CoNi@C – do the authors mean O₂ is dissociated on this surface? If so, what evidence is provided that O₂ is dissociated into O atoms in the absence of Pt? Perhaps the LEIS data provide this evidence, if the peak for O is characteristic of O atoms and not O₂.

9. Page 10 – Figure 3c – it was unclear what the model is that was used for Figure 3c – the Pt particle is much larger than the Pt₄ model used for DFT, Bader charge, and Pt adsorption energy calculations. This model may be more representative of the Pt particles present on the samples in this study.

10. SI comments:

Experimental section – a variety of details are missing that could inhibit efforts to reproduce the work.

a. Line 28: Please add the supplier for the SiO₂ spheres, or the procedure for synthesizing these.

- b. Line 29: For the equimolar Co and Ni solution what was the concentration used?
- c. Lines 30-32: For the reduction in H₂ and CH₄ to form CoNi, what flow rate per g catalyst was used, and what was the temperature ramp rate?
- d. Lines 34 and 40: what temperature and pressure were used for the vacuum drying steps?
- e. Line 38: how long was the ultrasonic oscillation performed?
- f. Line 57-58: additional details about the reaction chamber for LEIS should be added, or a reference to the design added.
- g. Line 60: what was the inner diameter and material used for the reactor? How was the reactor temperature measured (thermocouple inside the reactor or on the external surface?)
- h. Line 61: what was the particle size of the catalyst? Was the catalyst sieved before loading the reactor?
- i. Line 63: Can the authors add references or description of the "methane converter," and mention which GC column was used for separation?
- j. Line 64: What were the purities of the H₂, CO, O₂, and He used in the study?

11. Figure S3: It appears the catalyst would continue deactivating at 298 K if the reaction were run for >24 h. If possible, the authors should add an example of the reaction run for 72 h (or longer) without intermediate reduction steps, and then characterize the sample (XAS, TEM, XPS) afterwards to try to explain the cause of the deactivation. If there is no change in the CO pressure in the feed, why would the reaction rate decrease due to CO poisoning as a function of time on stream? In the caption of Figure S3: The last sentence of this caption should have a reference added, or additional description added.

Reviewer #2 (Remarks to the Author):

This is a study to understand the effect of graphene encapsulated transition metals (TM) on the activity of a supported metal NP-graphene interface. The authors conclude that the TM/graphene/TM configuration enhances the catalytic activity by the electron penetration effect from the CoNi alloy to Pt NP. The DFT calculations are used to support such results. I have major reservations.

1. To model the Pt-graphene interface, the authors employed a Pt₄ cluster model for the Pt nanoparticle (NP). Under this model, O₂ shows exclusive adsorption with Pt₄/CoNi@C compared to Pt₄/C. However, the finite-size effect in the simulation of NPs is critical and well-known to yield wrong binding energetics, especially for smaller sized clusters (1-5 atoms) (J. Phys. Chem. Lett. 2013, 4, 222–226 & Journal of Catalysis 223 (2004) 232–235). Experimental Pt NP is about 2 nm, but Pt₄ cluster used here to model it is ~0.5 nm. I strongly suspect the artifactual high binding affinity of this 4-atom cluster (of the order of > 1 eV according to the above refs).
2. The CoNi NP inside the carbon is also modelled by 55-atom cluster (1.2 nm in diameter) while experimental CoNi-NP is about 5 nm (~4500 atoms). Electronic structure of the model 55-atom cluster (1.2 nm) used here and realistic 4500-atom cluster (5 nm) are expected to be, again, quite different, and quite possibly a higher reactivity of smaller clusters may have contributed to the calculated stronger binding energetics claimed here.
3. If the electron penetration is important in Pt₄/CoNi@C, the local configuration of CoNi nanoparticle can be critical to affect the catalytic activity also. How did the authors determine the configuration of 55-atom CoNi alloy cluster (aside from the previous comment on the potentially artificial finite size effect of 55-atom cluster), and how the calculated energetics vary with different CoNi configurations?
4. In Fig.3, the adsorption energies are calculated without any CO coverages, but the reaction barriers are calculated with 4 pre-adsorbed CO. Why? The presence of pre-adsorbed CO molecules could affect greatly the adsorption energetics of CO, O₂, and the activation barriers altogether. One has to systematically show the coverage dependent energetics for CO binding, O₂ binding, and CO oxidation.

With all Pt atoms saturated with CO, O₂ could be adsorbed at the interface also.

5. To really show the effect of CoNi alloy inside the graphene, reaction profiles using Pt NP/CoNi@C should be compared with those using Pt NP/C.

Reviewer #3 (Remarks to the Author):

The manuscript reports enhanced CO oxidation reactivity of graphene-isolated Pt nanoparticles from CoNi alloy. The authors interpreted the enhanced reactivity, i.e. near 100 % CO conversion at room temperature, as modified Pt electronic structures due to electron penetration effect, i.e. delivering electrons from CoNi nanoparticles to Pt nanoparticle. With several state-of-art modern experimental tools and DFT calculation, the authors tried to deliver the messages that graphene-mediated electrons between Pt and CoNi nanoparticles play the major roles in the enhanced activity. While the DFT calculations shows energy states with detailed intermediate steps with convincing arguments, I cannot completely agree with the analysis of experimental results, which need to be clarified before the publication.

First, as a verification step, XANES was employed to estimate the d-hole of Pt nanoparticles. From the quality of figure, it is rather difficult to judge how the background subtraction was made for the calculation of d-hole. The slight change of XANES background normalization on higher photon energy side can easily change the estimated number of d-hole. It would be good to show the detailed figures in supplemental information for the clarification.

Second, the change of work function is suggested based on the DOS calculation. The change of work function can be probed with the use of NAP-XPS. The kinetic energy of gas phase signal can provide the modification of work function of the system. I wonder if the authors had a chance to look over this aspect from their results.

Lastly, but most importantly, according to the result of NAP-XPS, the graphene started to fail to protect CoNi nanoparticles from its oxidation at 150 C. To me, this shows that graphene is not fully attached to CoNi. Many previous reports showed that graphene on metal substrates can stay up to several hundred degrees of Celsius. The failure of protecting CoNi nanoparticles at this low temperature possibly indicate that there can be many defects on graphene layers. In fact, this graphene defects can easily trigger the activation of O₂ adsorption/dissociation. The role of graphene defects as active sites have been repeatedly reported and found to be very important for the study of graphene. I assume that most of the findings in this manuscript can be originated from the defect of graphene layers, instead of Pt-graphene interface.

If the authors can come up with the solid evidence of defect free graphene layer in this study, this manuscript can be reviewed again. At the moment, I cannot recommend the publication of this manuscript.

Response to the reviewers' comments

We are grateful for the reviewers' in-depth comments and suggestions, which have helped us a lot in improving the quality of this manuscript. The revised parts are highlighted in yellow in the files of "Revised Manuscript" and "Revised Supplementary Information".

Reviewer #1

Comments: This manuscript describes unique electron penetration effects from CoNi nanoparticles encapsulated by thin graphene layers, to Pt nanoparticles supported on the surface of the graphene. This complex material is capable of performing CO oxidation at 298 K, whereas carbon-supported Pt particles are not active until higher temperatures. Thorough evidence is included to demonstrate that CoNi are encapsulated (TEM, LEIS), and that the electronic state of Pt is altered by the presence of electron donation from metallic CoNi particles (through the graphene), namely XAS. I thoroughly enjoyed reading the manuscript and feel it presents a wide range of opportunities for future exploration, for those capable of synthesizing similar materials. It was unclear to me if the computational models employed accurately describe the as prepared or working catalyst, both due to the choice of a four-atom Pt cluster to represent 1-2 nm Pt nanoparticles. Additionally, the TEM images shows the possibility of multiple Pt particles adsorbing at a single encapsulated CoNi particle, but the impact of multiple Pt particles on the electron donation from CoNi was not included in the computational models. There are some spectroscopic data that were not included, but may have already been collected by the authors. Some of the experimental methods could be expanded to aid in reproduction of the research. The cause of slow deactivation at 298 K is unclear to me, and additional information could be added on this point. I describe these issues in more detail below. In summary, I applaud the authors on a thorough and intriguing study and recommend it for publication in Nature Communications after minor revisions.

Author Reply: Thank you very much for your positive evaluation on our manuscript. According to your useful suggestions, the experimental methods have been described in further detail, and more experiments and DFT calculations have been done. We would like to show those results one by one for each of your questions below.

Question 1. Page 3, Figure 1: Have the authors prepared Pt/(Co, Ni)/C samples, and how does their catalytic performance and spectroscopic characteristics compare to the Pt/CoNi@C materials? I would be interested to know how the d-hole count of Pt from XAS would look on a sample intentionally prepared to have Pt in contact with Co and/or Ni (e.g, perhaps bimetallic PtCo nanoparticles supported on C), as one would expect this might be different from the samples in this paper that have the C layer in between Pt and Co or Pt and Ni. This could provide further evidence that the authors proposed structure is present throughout the bulk of the sample (and not just in the limited regions observed by TEM), and if the spectroscopic data for PtCo particles on C is distinct from those reported in the manuscript, it would provide evidence that Co does not escape from the C shell and alloy with Pt over time. I will note that this is addressed to some extent on page 8, wherein PtCoNiO_x is less active than the fresh samples, which the authors propose results from a lack of free electrons in the CoNiO_x for donation to Pt.

Author reply: According to your suggestions, Pt-CoNiO_x/CB and Pt-CoNiO_x/CNT catalysts were intentionally prepared for comparison. First, CoNiO_x was precipitated on carbon supports in a form of layered double hydroxides with metals loading of 10 wt%. The procedure is the same as that for CoNiO_x/SiO₂ described in the section of Methods. Then, Pt NPs (1-2 nm) were deposited on CoNiO_x/CB and CoNiO_x/CNT with a Pt loading of 4 wt%. The procedure is the same as that for Pt/CoNi@NC described in the section of Methods. This two-step approach can create exposed Pt-CoNiO_x interfaces for a better comparison, because such interfaces but not PtCoNi alloy would be more likely formed for Pt/CoNi@NC catalyst if CoNi can escape from its graphene shell during CO oxidation reaction.

Fig. R1 shows the light-off curves of CO oxidation over Pt-CoNiO_x/CB and Pt-CoNiO_x/CNT catalysts, and the curves for Pt/CoNi@NC, Pt/CNT, and Pt/CB are from Fig. 2a of the manuscript. It can be seen that the catalytic activity is enhanced after introducing CoNiO_x onto both carbon supports and quite close for Pt-CoNiO_x/CB and Pt-CoNiO_x/CNT. The minor support effect indicated that most of Pt NPs should be in close contact with CoNiO_x on both catalysts. The formed Pt-CoNiO_x interfaces presented a much lower activity of CO oxidation than the fresh Pt/CoNi@NC catalyst (Fig. R1). It implies that the activity would decrease if Co or Ni escapes from the graphene shell during reaction, and this has been confirmed by the experimental design shown in Fig. 2d. However, the activity of the broken Pt/CoNi@NC catalyst (Fig. 2d) is a little lower than that of Pt-CoNiO_x/CNT (Fig. R1), which may be due to a lower number of Pt-CoNiO_x

interfaces for the former, because there are still some Pt NPs deposited on the concomitant CNT in Pt/CoNi@NC. Thus, it is better to do the comparison of XANES spectra between the fresh and broken Pt/CoNi@NC catalysts.

Fig. R1 Temperature-dependence CO conversion in CO oxidation reaction over the pre-reduced catalysts. 1% CO and 20% O₂ in He (1 bar). Space velocity: 60000 mL·g⁻¹·h⁻¹.

Fig. R2 shows the XANES spectrum of the broken Pt/CoNi@NC catalyst obtained after reacting to 240 °C (after the 2nd run in Fig. 2d), and the spectra of the fresh Pt/CoNi@NC catalyst and two control samples (Pt/CNT and Pt/CB) are from Fig. 1f of the manuscript. One can see that the jump height of Pt in Pt/CoNi@NC became higher after reacting to 240 °C and reached the same level as Pt in Pt/CNT, indicating a decreased d-band electron density. This data can support our inference that the oxidation of CoNi cannot enhance the catalytic activity significantly (the 3rd run in Fig. 2d) since CoNiO_x has no more free electrons to modulate the catalytic properties of the outer graphene and Pt NPs (Page 8). Thus, we would like to add Fig. R2 as Supplementary Fig. 12 in the SI. We also added Fig. R1 as Supplementary Fig. 11 with the above discussion and quoted it in the manuscript on Page 8. Thank you again for the valuable suggestions.

Fig. R2 XANES spectra of Pt L_{III} edge for Pt/CB, Pt/CNT, fresh and broken Pt/CoNi@NC.

Question 2. Page 3: The authors mention “it is interesting to find that Pt NPs are selectively deposited on the surface of graphene with CoNi NPs inside but rarely on the surface of the concomitant CNT” and refer the reader to TEM images. The TEM images will only represent a minority of each sample, is there any additional quantitative surface characterization the authors can provide that Pt NPs are selectively deposited near the CoNi NPs? Perhaps IR spectra of adsorbed CO (or Pt XPS) would show a different peak center for Pt/C compared to Pt near encapsulated CoNi? The XANES data show, on average, that Pt/CoNi@C has different edge energy than Pt/C, but these data for Pt/CoNi@C could reflect a mixture of Pt particles distant from CoNi particles and Pt particles adsorbed at CoNi.

Author Reply: As described in the last paragraph of Page 3 of the manuscript, we have two samples of graphene encapsulating CoNi alloy, CoNi@NC and CoNi@C, and CNT is concomitantly generated only in CoNi@NC but not in CoNi@C. Thus, for Pt/CoNi@NC, the XANES data reflect the average information of the electronic structure of all the Pt NPs with different locations (either on the surface of CNT or on the surface of graphene with CoNi inside). While for Pt/CoNi@C, the XANES data (orange dash line in Fig. 1f) can exclusively reflect the electronic structure of Pt on the surface of graphene with CoNi inside. Then, through comparing the d-hole count of Pt in Pt/CoNi@NC, Pt/CoNi@C and Pt/CNT calculated from the XANES spectra (Fig. 1g) we can estimate that more than 80% of Pt NPs in Pt/CoNi@NC are deposited on the surface of graphene with CoNi inside.

CO-adsorbed IR spectroscopy, as a surface-sensitive technique, is usually the first choice for in-situ characterization for CO oxidation reaction, especially in our electron-mediated situation. We had tried the CO-adsorbed IR testings over Pt/CoNi@NC and other catalysts in either DRIFTS or transmission mode even at a low temperature ($-40\text{ }^{\circ}\text{C}$), but no signal of adsorbed CO can be observed after slightly vacuuming (Fig. R3a) or simply purging with Ar (Fig. R3b). It should be due to the strong absorption of IR on such a black material, which is a common phenomenon for carbon-based catalysts.

Fig. R3 CO-adsorbed IR spectra over Pt/CoNi@NC in (a) transmission mode and (b) DRIFTS.

Question 3. Page 5 and Figure 1: is the Pt_4 model realistic given that the Pt NPs are 1-2 nm in size and likely comprised of >4 Pt atoms? This is my strongest criticism of the paper in multiple areas – whether or not this model is representative of the samples. Were other models with larger numbers of Pt atoms considered, and were the results similar to those reported here? The effect of the number of Pt atoms on the calculated parameters should be included in the article or the SI. Additionally, the authors model a single Pt_4 particle on each CoNi cluster, but from TEM (e.g., Figure 1d) it appears plausible that multiple Pt particles are attached to each encapsulated CoNi cluster. What is the effect of inclusion of multiple Pt particles on the calculated parameters? I would imagine either a larger number of atoms per particle, or a larger number of Pt particles per CoNi cluster, would disperse the electronic effects to a greater extent, perhaps minimizing the magnitude of the difference relative to C without CoNi encapsulated. These calculated parameters include Bader charge and Pt binding energy (page 5, Figure 1h), PDOS (Figure 1j), DFT calculations (Figure 3).

Author Reply: According to your suggestions, we appended two systems to investigate the size effect of Pt cluster. One is to increase the number of Pt atoms from 4 to 9 (Fig. R4), and the other is to build Pt nano-strip (81 atoms) on the graphene layer with or without CoNi underneath (Fig. R5). The results of Bader charge analysis and the binding energies between Pt and graphene are listed in Table R1. For Pt₉, it possesses a negative charge on CoNi@C but a positive charge on C. For Pt nano-strip, it also possesses a negative charge on graphene/CoNi but a positive charge on graphene. Both trends are the same as for Pt₄ models used in our manuscript. The binding energy between Pt and C also shows the same trend for those models, which can be enhanced after introducing CoNi NPs. The lower binding energy of Pt₉ on C than that of Pt₄ on C is due to the increase of cohesive energy of Pt cluster for a larger cluster. The higher binding energy of Pt nano-strip models than others is due to the larger number of Pt atoms. The PDOS analysis for those systems also gives the same trend of increasing the Fermi level of C after introducing CoNi NPs, as shown in Fig. R6.

Fig. R4 Structures of Pt₉ cluster supported on (a) CoNi@C and on (b) C.

Fig. R5 Structures of Pt nano-strip on (a) graphene/CoNi and on (b) graphene from side view and top view.

Table R1. Bader charge of Pt cluster and binding energy between Pt and C on different models.

Model	Pt ₄		Pt ₉		2Pt ₄		Pt nano-strip	
	CoNi@C	C	CoNi@C	C	CoNi@C	C	CoNi@C	C
Charge	-0.08	+0.15	-0.09	+0.13	-0.12	+0.26	-0.32	+0.80
E _b	-2.74	-2.17	-2.74	-1.38	-5.50	-4.24	-28.66	-18.85

**Fig. R6** Comparison of PDOS of 2s+2p orbitals of C atoms between (a) Pt₄/CoNi@C and Pt₄/C, (b) Pt₉/CoNi@C and Pt₉/C, (c) 2Pt₄/CoNi@C and 2Pt₄/C, and (d) Pt_{strip}/CoNi@C and Pt_{strip}/C.

According to your suggestions, two Pt₄ clusters supported on CoNi@C (Fig. R7) were constructed to simulate multiple Pt NPs on one encapsulated CoNi NPs. As seen from the data of Bader charge and binding energy (Table R1) and the PDOS of C (Fig. R6), the overall trend

does not change for two Pt₄ clusters comparing with one Pt₄ cluster. It is reasonable in consideration of that the electron penetration from CoNi to Pt through graphene should mainly occur at the local interface. We appended all the models and calculation results here to the SI and quoted them in the manuscript on Page 5.

Fig. R7 Structures of two Pt₄ clusters supported on (a) CoNi@C and on (b) C.

Question 4. Page 5 and Figure 1h: is Pt binding and Bader charge at an empty C sphere an appropriate comparison for these characteristics when Pt binds to C-encapsulated CoNi particles? When not binding to an encapsulated CoNi, Pt would realistically be bound to an extended C surface, so perhaps a filled C sphere would be a better comparison for these calculations than a hollow C sphere (assuming there are not hollow C spheres in the C supports used by the authors).

Author Reply: According to your suggestions, we compared the electronic structure of Pt₄ cluster on an empty C sphere with that on a filled C sphere (Fig. R8). The difference of Bader charge of Pt₄ cluster between the two models is about 0.006 e⁻, which is neglectable. The reason is that the van der Waals interaction between the carbon layers is too weak to affect the strong bonding interaction between Pt and C.

Fig. R8 Structure of Pt₄ cluster on filled C sphere. The inner carbon sphere is colored in brown.

Question 5. Page 6: what leads to the deactivation over 24 h, and why can the material be regenerated in H₂? Have the authors performed XPS (or TEM) on the catalyst after 24 h reaction, and how does this compare to that for the fresh catalysts? TEM after exposure to reaction conditions at higher temperatures is reported in Figure 2c, but I did not see the TEM for the materials after 24 h exposure to reaction conditions at 298 K.

Author Reply: According to your suggestions, we supplemented the TEM and XPS characterizations on the sample of Pt/CoNi@NC after reacting at 25 °C for 24 h. As shown in Fig. R9, the graphene layer of the used catalyst is still intact and the inner CoNi is still in the metallic state. It indicates that the active sites are still the same as the fresh catalyst (both Pt-graphene interfaces and Pt-Pt pairs). Thus, the gradual deactivation should be attributed to the gradual decrease in the number of active sites. Although CO has a stronger adsorption on Pt than O₂ does (Fig. 3a), in the initial stage of the reaction, CO with only 0.01 bar (even much lower at such a high conversion close to 100%) may not fully cover the Pt sites in consideration of the high partial pressure of O₂ (0.2 bar). Then, along with the time of stream, CO will poison some Pt sites (mainly Pt-Pt pairs) and cause the deactivation. It is reasonable because of the negative reaction order for CO (-1) and the positive reaction order for O₂ (1) over Pt catalyst (J. Am. Chem. Soc. 2011, 133, 4498-4517), and this CO poisoning mechanism were also used to explain the deactivation of CO oxidation over one Pd catalyst (Angew. Chem. Int. Ed. 2015, 54, 15823-15826). However, a possible O₂-induced deactivation can still not be excluded in our case, as the electron-rich Pt atoms especially around the interfaces may be more susceptible to oxidation under such a high partial pressure of O₂ and the oxidized Pt will present a lower activity for CO oxidation than the metallic Pt (J. Phys. Chem. C 2016, 120, 17996-18004). Actually, we have got a preliminary experimental hint for this possibility, that is, the pre-reduced Pt/CoNi@NC catalyst after purging with 20% O₂/Ar at room temperature for a while exhibited a lower activity for CO oxidation. This phenomenon is unusual for CO oxidation over Pt-based catalysts (e.g. Pt/CNT). A thorough kinetic study is undergoing to investigate the deactivation mechanism. In any case, the decrease in the number of active sites caused by either CO poisoning or O₂ oxidation can be easily restored by a simple H₂ reduction. We added Fig. R9 as Supplementary Fig. 8 in the SI and quoted it in the manuscript on Page 6. Thank you for this useful suggestion.

Fig. R9 TEM image of Pt/CoNi@NC after reaction at room temperature for 24 h and the corresponding XPS spectra comparing with the freshly-reduced catalyst.

Question 6. Page 6: what was observed for Pt during the NAP-XPS experiments? It would benefit the reader if the Pt XPS data during these treatments were included in Figure 2. Furthermore, I would note that these CO and O₂ pressures are much lower than those used for catalysis, so the effect of extended exposure to 1 kPa CO and 20 kPa O₂ on the oxidation states of Pt, Co, and Ni at 298 K is unclear from the data reported in Figure 2b. Could the authors perform operando Pt, Co, and/or Ni XAS to monitor the oxidation states of these elements at reaction conditions?

Author Reply: As seen from Fig. R10, there is no obvious difference for Pt 4f_{7/2} signal at varied temperatures (25-210 °C) during the NAP-XPS testings. A Cu sample stage was used to hold the catalyst, and the Cu 3p signal overlapped the Pt 4f_{5/2} signal. Thus, the Pt 4f_{5/2} signal is not shown here. We added Fig. R10 as Supplementary Fig. 9 in the SI and quoted it in the manuscript on Page 7.

Fig. R10 XPS spectra of Pt $4f_{7/2}$ from NAP-XPS testings over Pt/CoNi@NC in a flow of 0.067 mbar CO and 1.13 mbar O_2 at different reaction temperatures (25-210 °C). The reduction process was performed after reaction.

We are sorry that we cannot make the operando XAS experiments due to the restrictions of platform facilities and booking time, but a quasi-in-situ XPS testing was performed over Pt/CoNi@NC to observe the change of valence states of both metals (Pt, Co, and Ni) and light elements (C, O, and N). Same to the LEIS setup, we connected the in-situ reaction chamber next to the UHV-XPS chamber. The in-situ reaction chamber allows the reduction and reaction of catalysts with the same parameters of temperature and pressure as for the actual testings. Firstly, the sample was pretreated in a flow of H_2 at 240 °C for 2 h. Then, the reaction was performed under a flow of mixed gas (1% CO and 20% O_2 in He) at room temperature for 72 h. After each 24 h, the in-situ reaction chamber was evacuated to UHV, and then the sample was transferred into the UHV-XPS chamber for detection. As shown in Fig. R11, there is no significant change for Co, Ni, Pt, and N signals, but the O signal gradually increases after each reaction stage (Table R2) and the peak position of 531.5 eV corresponds to the C=O species, indicating the activation of O_2 on the graphene surface.

Fig. R11 XPS spectra of C 1s, N 1s, O 1s, Co 2p, Ni 2p, and Pt 4f from in-situ XPS testings over Pt/CoNi@NC. 25 °C, 1% CO and 20% O₂ in He (1 bar). Space velocity: 60000 mL·g⁻¹·h⁻¹.

Table R2. Atomic ratios of N/C and O/C obtained from Fig. R11.

atomic ratio	raw	reduced	24 h	48 h	72 h
N/C	3.1%	3.0%	2.9%	3.2%	3.1%
O/C	2.0%	1.5%	2.2%	2.6%	3.3%

Question 7. Page 7, Figure 2: for the caption of figure 2e, it would help if the treatment conditions were included in the caption (gas pressures, treatment durations, flowrates per mass catalyst, etc.).

Author Reply: Thank you for the kind reminder. We added the reaction conditions (1% CO and 20% O₂ in He (1 bar). Space velocity: 60000 mL·g⁻¹·h⁻¹) in the caption of Fig. 2e.

Question 8. Page 9: Figure 2e shows that there is O on the samples after reaction at 298 K, that cannot be removed even after reduction at 513 K. The authors propose this O reflects activation of O₂ on the graphene surface. Is this signal O₂, or O adatoms? In either case, the fact that these are not removed during after reduction at 240 °C suggests that these may be structural or strongly bound O atoms, that may not be participating in catalysis. Are these present on the CoNi@C and C materials in the absence of Pt? It was unclear to me if these O are part of the same reservoir of O that is ultimately incorporated in CO₂, or if these O are spectators/unreactive. In the caption of Figure S5, the authors mention that O₂ can be activated on the surface of CoNi@C – do the authors mean O₂ is dissociated on this surface? If so, what evidence is provided that O₂ is dissociated into O atoms in the absence of Pt? Perhaps the LEIS data provide this evidence, if the peak for O is characteristic of O atoms and not O₂.

Author Reply: Thank you for the insightful comment. For LEIS, ultra-high vacuum is needed to evacuate the chamber before testing. Thus, the detected signal of O corresponds to the O atoms strongly bound to the surface of catalyst but not the O₂ molecule. Therefore, the increased O signal after reaction (Fig. 2e) indicated the dissociation of O₂ on the catalyst. According to your suggestion, here we performed the LEIS testings over CoNi@NC and CB without loading Pt. To avoid some potential influence of CO, synthetic air was used to flow through the samples at room temperature for 2 h between two reduction processes (H₂, 240 °C, and 2 h). As seen from Fig. R12, the O signal of CoNi@NC increased after exposing in air flow and cannot be efficiently removed by such a mild reduction, which is consistent with the results of Pt/CoNi@NC shown in Fig. 2e. In contrast, the O signal of CB did not change too much after these treatments (Fig. R12). As a conclusion, O₂ can be activated on CoNi@NC but not on CB at room temperature, which can further confirm that the graphene turns to be more active when encapsulating CoNi NPs. We added Fig. R12 as Supplementary Fig. 13 to further support this conclusion on Page 9.

Fig. R12 LEIS spectra of CoNi@NC and CB after exposing in air and reduction in H₂.

As we stated on Page 9, considering that CO needs to be activated on Pt NPs, only the activated O on graphene adjacent to the periphery of Pt NPs can participate in the CO oxidation reaction. Therefore, we suppose that the Pt-graphene interface should be the main active site for CO oxidation. Whether the remote O far from the interface can participate in the reaction probably depends on the diffusion barrier of O on the graphene surface, and this may not happen in consideration of the strong covalent bond between O and C.

Question 9. Page 10 – Figure 3c – it was unclear what the model is that was used for Figure 3c – the Pt particle is much larger than the Pt₄ model used for DFT, Bader charge, and Pt adsorption energy calculations. This model may be more representative of the Pt particles present on the samples in this study.

Author Reply: The model in Fig. 3c is a schematic diagram and was not used in our calculations. This model is indeed more representative, but the computational cost is too high to even perform static calculations, let alone the reaction-related calculations shown in Fig. 3b. It contains more than 1000 carbon atoms and 1000 metal atoms, which is very challenging for the DFT method. However, as mentioned above, we constructed a similar Pt nano-strip model (Fig. R5) with 280 carbon atoms and 301 metal atoms for comparison, and confirmed that our Pt₄/CoNi@C model is sufficient to reveal the active site and reaction mechanism by DFT calculation with a balance between accuracy and efficiency.

Question 10. *SI comments: Experimental section – a variety of details are missing that could inhibit efforts to reproduce the work.*

- a. Line 28: Please add the supplier for the SiO₂ spheres, or the procedure for synthesizing these.*
- b. Line 29: For the equimolar Co and Ni solution what was the concentration used?*
- c. Lines 30-32: For the reduction in H₂ and CH₄ to form CoNi, what flow rate per g catalyst was used, and what was the temperature ramp rate?*
- d. Lines 34 and 40: what temperature and pressure were used for the vacuum drying steps?*
- e. Line 38: how long was the ultrasonic oscillation performed?*
- f. Line 57-58: additional details about the reaction chamber for LEIS should be added, or a reference to the design added.*
- g. Line 60: what was the inner diameter and material used for the reactor? How was the reactor temperature measured (thermocouple inside the reactor or on the external surface?)*
- h. Line 61: what was the particle size of the catalyst? Was the catalyst sieved before loading the reactor?*
- i. Line 63: Can the authors add references or description of the “methane converter,” and mention which GC column was used for separation?*
- j. Line 64: What were the purities of the H₂, CO, O₂, and He used in the study?*

Author Reply: Thank you for your kind reminder. We refined the description of the Methods section and moved it from the SI to the Manuscript file. Please check it from Page 11. For your questions here, our responses are attached below.

- a. The 100-nm SiO₂ sphere colloidal dispersion was purchased from Alfa Aesar. The stock number is 43094.
- b. 1.06 g of Co(NO₃)₂·6H₂O and 0.90 g of Ni(CH₃COO)₂·4H₂O were dissolved into 100 mL of water for use. Thus, the concentrations of Co²⁺ and Ni²⁺ are both 0.036 mol/L.
- c. 200 mg of CoNi(OH)_x/SiO₂ was heated from 30 to 700 °C at a rate of 10 °C/min in a 100 mL/min of 30% H₂/Ar to obtain CoNi metallic alloy. After reaching 700 °C, H₂ was switched off and CH₃CN (bubbling with 70 mL/min of Ar) or CH₄ (100 mL/min, without Ar) was introduced as carbon sources to form graphene on CoNi NPs.
- d. The graphene encapsulating CoNi alloy samples including those loading Pt NPs were dried at 40 °C in vacuum (~0.1 kPa) for 12 h.
- e. The ultrasonic oscillation was performed for 0.5 h.

- f. The reaction chamber was actually modified from a tube furnace. The sample holder for LEIS measurement can be transferred into the isothermal zone of the furnace for reaction.
- g. CO oxidation was carried out in a quartz tube (6 mm outer diameter and 4 mm inner diameter) reactor. Temperature was measured with a thermocouple on the external surface of quartz tube right next to the catalyst bed.
- h. Catalysts were sieved through a 200-mesh sieve so that the particle size is within 80 μm .
- i. The methane converter is a micro high-temperature furnace (330 $^{\circ}\text{C}$) with Ni-based catalyst that can fully convert CO and CO₂ into CH₄ for FID detection. Instead of TCD, FID was used to achieve a higher sensitivity for GC analysis. Thus, the methane converter is necessary considering that CO and CO₂ cannot be directly detected by FID. The GC column used here is a packed column (TDX-01).
- j. The purity of H₂ and Ar is 99.999%. We didn't use the pure gases of CO, O₂ and He but a mixed gas of 1% CO, 20% O₂, and 79% He as the reaction gas for this work.

Question 11. *Figure S3: It appears the catalyst would continue deactivating at 298 K if the reaction were run for >24 h. If possible, the authors should add an example of the reaction run for 72 h (or longer) without intermediate reduction steps, and then characterize the sample (XAS, TEM, XPS) afterwards to try to explain the cause of the deactivation. If there is no change in the CO pressure in the feed, why would the reaction rate decrease due to CO poisoning as a function of time on stream? In the caption of Figure S3: The last sentence of this caption should have a reference added, or additional description added.*

Author Reply: According to your suggestion, we performed the 72-h stability testings over Pt/CoNi@NC at 25 $^{\circ}\text{C}$ under two space velocities (Fig. R13). The initial CO conversion can also reach near 100% under a shorter residence time (the higher space velocity), and the ratio of CO conversion between the two space velocities tends to approach 2 in the final stage. It means that the coverages of CO and O₂ have not reach the steady state at the initial stages with high CO conversions for both curves, indicating a CO poisoning process occurred. Also from Fig. R13, we can observe two deactivation rates with 36 h as dividing line, indicating a different deactivation mechanism existed. As stated in the reply to your Question 5, a possible O₂-induced deactivation may co-exist. At present, by only judging from the TEM and XPS data (Fig. R9 and Fig. R11) the deactivation mechanism is still unclear. A more sensitive operando XAS (in plan) may provide

more information and an in-depth kinetic study is more necessary and undergoing for this purpose, which hope to form another independent work. Here for the description in the caption of Supplementary Fig.7 (Figure S3 in previous SI), it was modified as “the slow deactivation should be due to either the gradual CO poison on Pt NPs or the possible oxidation of Pt NPs, which is still under investigation”.

Fig. R13 Long-term stability of Pt/CoNi@NC towards CO oxidation at 25 °C for 72 h under a mixed flow of 1% CO and 20% O₂ in He (1 bar) with two total space velocities as indicated.

Reviewer #2

Comments: This is a study to understand the effect of graphene encapsulated transition metals (TM) on the activity of a supported metal NP-graphene interface. The authors conclude that the TM/graphene/TM configuration enhances the catalytic activity by the electron penetration effect from the CoNi alloy to Pt NP. The DFT calculations are used to support such results. I have major reservations.

Question 1. To model the Pt-graphene interface, the authors employed a Pt₄ cluster model for the Pt nanoparticle (NP). Under this model, O₂ shows exclusive adsorption with Pt₄/CoNi@C compared to Pt₄/C. However, the finite-size effect in the simulation of NPs is critical and well-known to yield wrong binding energetics, especially for smaller sized clusters (1-5 atoms) (*J. Phys. Chem. Lett.* 2013, 4, 222–226 & *Journal of Catalysis* 223 (2004) 232–235). Experimental Pt NP is about 2 nm, but Pt₄ cluster used here to model it is ~0.5 nm. I strongly suspect the artifactual high binding affinity of this 4-atom cluster (of the order of > 1 eV according to the above refs).

Author Reply: Thank you for your insightful comment. Smaller sized clusters usually present higher adsorption energy, which is a very common phenomenon for DFT calculations. We also noticed the relatively high adsorption energies for both O₂ and CO on our model than on the Pt slab model reported elsewhere (e.g. Science 2010, 328, 1141-1144). Thus, we appended a larger Pt nano-strip model (Fig. R14) to simulate the interface structure in our real catalyst. It consists of 81 Pt atoms, 110 Co atoms, 110 Ni atoms and 280 carbon atoms. The electronic structure analysis (Table R3 and Fig. R15) indicates that, despite the quantitative values are different, the main trend and conclusion are the same for Pt₄ cluster model and Pt nano-strip model. For instance, the charge density of Pt is increased and the work function of graphene is decreased when introducing CoNi into each model. Furthermore, the binding strength for Pt on graphene/CoNi is also stronger than that on graphene, which can be directly observed from the side view of the structures (Fig. R14).

Fig. R14 Structures of Pt nano-strip on (a) graphene/CoNi and on (b) graphene from side view and top view.

Table R3. Bader charge of Pt cluster and binding energy between Pt and C on different models.

Model	Pt ₄		Pt nano-strip	
	CoNi@C	C	CoNi@C	C
Charge	-0.08	+0.15	-0.32	+0.80
E _b	-2.74	-2.17	-28.66	-18.85

Fig. R15 Comparison of PDOS of 2s+2p orbitals of C atoms between Pt₄/CoNi@C and Pt₄/C (left), and Pt_{strip}/CoNi@C and Pt_{strip}/C (right).

Based on the Pt nano-strip model, we further calculated the adsorption energy of CO and O₂ on Pt or at the Pt-graphene interface (Fig. R16 or Supplementary Fig. 17). Comparing with Pt₄ cluster models (Table R4), the adsorption of CO on Pt nano-strip models become slightly weaker but are still stronger than those reported in literatures (e.g. -1.64 eV for CO on Pt(111) from the above mentioned paper), which is mainly because the adsorption in the literatures took place at close-packed sites whereas edge sites were used here. However, the main conclusion of CoNi-enhanced CO adsorption drawn from the P₄ cluster model is still valid for the Pt nano-strip model.

Fig. R16 Optimized structures for adsorption of (a,b) CO and (c,d) O₂ on Pt nano-strip models with and without CoNi underneath the graphene.

Table R4. Adsorption energies of CO and O₂ on Pt or at interface in different models.

Adsorption energy (eV)	Pt ₄		Pt nano-strip	
	CoNi@C	C	CoNi@C	C
CO on Pt	-2.77	-2.57	-2.40	-2.21
O ₂ on Pt	-1.18	-0.98	-1.80	-1.62
O ₂ at interface	-0.37	No adsorption	-1.21	No adsorption

For the adsorption of O₂, same to the Pt₄ cluster model, we considered two adsorption sites, Pt-graphene interface site and Pt site, on the Pt nano-strip models. For the interface site, there is still no stable adsorption configuration for the model without CoNi alloy. After introducing CoNi, O₂ can adsorb at the Pt-graphene interface with an even higher binding energy comparing to the Pt₄ cluster model, which means it is more favorable for O₂ adsorption on a larger model. In short, the Pt-graphene interface provides the site for the activation of O₂ benefiting from the encapsulated CoNi. This is one of the main conclusions of our work, which can be proved by both models.

Above all, the conclusions from Pt₄ model were further validated by the more representative Pt nano-strip model. On the other side, Pt nano-strip model contains about 600 atoms (Pt₈₁, Co₁₁₀, Ni₁₁₀, and C₂₈₀), which is still a challenge for efficiently calculating the reaction mechanism at a reliable accuracy. There are also other technical issues, such as spin-polarization for Co and Ni elements, lattice mismatch, which further decreases the efficiency. The model of Pt₄/CoNi@C is trying to achieve a balance between accuracy and efficiency.

Question 2. *The CoNi NP inside the carbon is also modelled by 55-atom cluster (1.2 nm in diameter) while experimental CoNi-NP is about 5 nm (~4500 atoms). Electronic structure of the model 55-atom cluster (1.2 nm) used here and realistic 4500-atom cluster (5 nm) are expected to be, again, quite different, and quite possibly a higher reactivity of smaller clusters may have contributed to the calculated stronger binding energetics claimed here.*

Author Reply: This issue can be addressed by the above mentioned Pt nano-strip model, where CoNi NP was constructed by CoNi alloy (111) slab model. In this model, the CoNi surface is close-packed and is usually considered as the least active surface for reaction. However, according to above discussion, the conclusions from CoNi-55 spherical model were further

validated by the more representative CoNi-220 slab model. All the appended models and calculation results were attached in the SI (Page 5-6).

Question 3. *If the electron penetration is important in Pt₄/CoNi@C, the local configuration of CoNi nanoparticle can be critical to affect the catalytic activity also. How did the authors determine the configuration of 55-atom CoNi alloy cluster (aside from the previous comment on the potentially artificial finite size effect of 55-atom cluster), and how the calculated energetics vary with different CoNi configurations?*

Author Reply: In the Pt₄/CoNi@C model, the configuration of CoNi is randomly generated in order to simulate the randomly distributed alloy. To address the effect of CoNi configuration, in principle, it needs large number sampling. We used a simpler method here, that is, to reverse the order of Co and Ni to simulate the opposite configuration. According to the results listed in Table R5, the charge of Pt cluster can also gain electrons (0.06 e⁻) from the reversed CoNi. CO adsorption is slightly weakened by 0.11 eV, whereas O₂ adsorption at interface is enhanced from -0.37 to -0.75 eV. The data further validate the enhancement of electron penetration effect on the reaction, and the trend becomes more favorable. It is reasonable to predict that the average effect falls in between these two configurations if we take more configurations into account.

Table R5. Adsorption energies of CO on Pt and O₂ at Pt-graphene interface and the Barder charge of Pt₄ cluster in different models.

	Pt ₄ /CoNi@C	Pt ₄ /reversed-CoNi@C	Pt ₄ /C
CO on Pt (eV)	-2.77	-2.66	-2.57
O ₂ at interface (eV)	-0.37	-0.75	No adsorption
Barder Charge of Pt ₄	-0.08	-0.06	0.15

Question 4. *In Fig.3, the adsorption energies are calculated without any CO coverages, but the reaction barriers are calculated with 4 pre-adsorbed CO. Why? The presence of pre-adsorbed CO molecules could affect greatly the adsorption energetics of CO, O₂, and the activation barriers altogether. One has to systematically show the coverage dependent energetics for CO binding, O₂ binding, and CO oxidation. With all Pt atoms saturated with CO, O₂ could be adsorbed at the interface also.*

Author Reply: Thank you for the valuable suggestion. We calculated the adsorption energy of CO on both Pt₄/CoNi@C and Pt₄/C with different coverages, and the results are shown in Fig. R17. Here, the adsorption energy of 4CO and 5CO refers to the energy of the 4th and 5th CO adsorbing on the 3CO-adsorbed and 4CO-adsorbed Pt₄ clusters, respectively. 5CO can reach the saturated coverage on our Pt₄ cluster. It can be seen that the adsorption of CO on both Pt₄/CoNi@C and Pt₄/C will be weakened along with increasing the coverage of CO. The same rule applies for the adsorption of O₂. For example, the adsorption energy of O₂ at the Pt-graphene interfaces of Pt₄/CoNi@C decreases from -0.37 to -0.21 eV when saturating with 5CO on Pt₄ cluster (Table R6). However, the adsorption of the 5th CO on 4CO-adsorbed Pt₄ (around -1.6 eV) is still stronger than that of O₂ on a clean Pt₄ (around -1 eV, Fig. 3a), indicating that O₂ cannot compete with CO for adsorption on Pt site no matter with the coverage of CO. That is why we start the energy profile from 4CO-adsorbed Pt₄ (model I in Fig. 3b), and the first step is the adsorption of the 5th CO on it to reach the saturated coverage. After that, we found that O₂ can still adsorb at the Pt-graphene interface of Pt₄/CoNi@C (model III, $\Delta E = -0.21$ eV) but cannot adsorb at the interface of Pt₄/C, as we described on Page 11 of the manuscript. For the influence of CO coverage on the reaction barrier, we calculated the combination step between adsorbed CO and O on a clean Pt₄/CoNi@C, which gave a slightly lower barrier of 0.47 eV relative to that of 0.63 eV on the CO-saturated Pt₄/CoNi@C (Fig. 3b). We added Fig. R17 as Supplementary Fig. 15 in the SI and quoted it in the manuscript on Page 9.

Fig. R17 Adsorption energy of CO on Pt₄/CoNi@C and Pt₄/C with different coverages.

Table R6. Adsorption energies of CO and O₂ and reaction barrier between CO and O on clean Pt₄ and 4CO-adsorbed Pt₄ models.

Energy (eV)	Clean Pt ₄		4CO-adsorbed Pt ₄	
	CoNi@C	C	CoNi@C	C
CO on Pt	-2.77	-2.57	-1.59	-1.67
O ₂ at interface	-0.37	No adsorption	-0.21	No adsorption
Barrier of CO+O→CO ₂	0.47	-	0.63	-

Question 5. To really show the effect of CoNi alloy inside the graphene, reaction profiles using Pt NP/CoNi@C should be compared with those using Pt NP/C.

Author Reply: We had tried to systematically compare the full reaction profile between Pt₄/CoNi@C and Pt₄/C. However, the step of O₂ adsorption at the Pt-graphene interface in Pt₄/C is not possible since no local minimum exists for this configuration. Therefore, the subsequent reaction is almost impossible through this mechanism. We proposed that the reaction for Pt₄/C follows the classical mechanism, where O₂ adsorption has to compete with CO adsorption on Pt site. This will result in low activity at room temperature for Pt/C catalyst. On the other hand, as shown in Fig. 3b of the manuscript, the free energy of O₂ adsorption for Pt₄/C model is even higher than the maximum barrier of the whole reaction profile for Pt₄/CoNi@C model, which can already illustrate the significant role of the encapsulated CoNi.

Reviewer #3

Comments: The manuscript reports enhanced CO oxidation reactivity of graphene-isolated Pt nanoparticles from CoNi alloy. The authors interpreted the enhanced reactivity, i.e. near 100 % CO conversion at room temperature, as modified Pt electronic structures due to electron penetration effect, i.e. delivering electrons from CoNi nanoparticles to Pt nanoparticle. With several state-of-art modern experimental tools and DFT calculation, the authors tried to deliver the messages that graphene-mediated electrons between Pt and CoNi nanoparticles play the major roles in the enhanced activity. While the DFT calculations shows energy states with detailed intermediate steps with convincing arguments, I cannot completely agree with the analysis of experimental results, which need to be clarified before the publication.

Question 1. First, as a verification step, XANES was employed to estimate the d-hole of Pt nanoparticles. From the quality of figure, it is rather difficult to judge how the background subtraction was made for the calculation of d-hole. The slight change of XANES background normalization on higher photon energy side can easily change the estimated number of d-hole. It would be good to show the detailed figures in supplemental information for the clarification.

Author Reply: Thank you for your suggestion. Fig. R18 shows the original XAS spectra of Pt/CoNi@NC, Pt/CoNi@C, Pt/CNT, Pt/CB, Pt foil, and PtO₂. Their edge positions are exactly the same. Thus, during processing the data, E₀ was firstly calibrated to 11564.0 eV with the same energy shift of -0.1 eV for all the spectra. Also from Fig. R18, the absorption strength is weaker for the four catalysts samples than Pt foil and PtO₂ due to their low Pt loadings. Thus, the pre-edge and post-edge ranges were initially determined based on the spectra of Pt foil and PtO₂ and then applied to others to ensure the same ranges for all the samples. The pre-edge range we used is from -180 to -80 eV relative to E₀, and the post-edge range is from +150 to +800 eV relative to E₀. From Fig. R19, we can see that these ranges can fit all the samples well. The spectrum of Pt/CoNi@C has a slight bend at the high energy region probably due to the background drift. We have added Fig. R19 as Supplementary Fig. 3 in the SI according to your suggestion. Actually, if we use a pre-edge range of -150 to -50 eV relative to E₀, the XANES spectra (Fig. R20a) are almost unchanged comparing with Fig. 1f, and so do the estimated d-hole counts. If we change the post-edge upper limit from +800 to either +750 or +850 eV, only the spectrum of Pt/CoNi@C fluctuates a lot (Fig. R20, b and c), but this fluctuation does not change the calculated d-hole counts (1.6 ± 0.02) of this sample too much.

Fig. R18 Original XAS spectra of Pt foil, PtO₂, Pt/CoNi@NC, Pt/CoNi@C, Pt/CNT, and Pt/CB.

Fig. R19 XAS spectra of Pt foil, PtO₂, Pt/CoNi@NC, Pt/CoNi@C, Pt/CNT, and Pt/CB with pre-edge (−180 to −80 eV) and post-edge (+150 to +800 eV) baselines before normalization.

Fig. R20 XANES spectra of all samples normalized with different pre-edge and post-edge ranges.

Question 2. Second, the change of work function is suggested based on the DOS calculation. The change of work function can be probed with the use of NAP-XPS. The kinetic energy of gas phase signal can provide the modification of work function of the system. I wonder if the authors had a chance to look over this aspect from their results.

Author Reply: Thank you for this useful suggestion. We checked the O 1s XPS spectra from the NAP-XPS testings over Pt/CoNi@NC and Pt/CNT in a flow of 0.067 mbar CO and 1.13 mbar O₂ at room temperature, and the results are shown in Fig. R21. The two samples were loaded on two regions of the same sample holder, enabling a high comparability. The peak at around 539 eV in Fig. R21 is attributed to the signals from CO and O₂ gases. The doublet of O₂ merged together, and the CO signal may also merge into the whole peak if it can be detected. However, we can still observe that this peak of gas phase over Pt/CoNi@NC has a clear blue shift of 0.3 eV relative to that over Pt/CNT. As proposed by Liu et al. (Nano Lett. 2013, 13, 6176-6182), there is a negative correlation between the work function of material surface and the binding energy of gas phase over it. Thus, this data experimentally proved the decrease of work function of graphene in Pt/CoNi@NC. We added it as Supplementary Fig. 6 in the SI to support our DFT calculation on Page 6 of the manuscript.

Fig. R21 XPS spectra of O 1s from NAP-XPS testings over Pt/CoNi@NC and Pt/CNT in a flow of 0.067 mbar CO and 1.13 mbar O₂ at room temperature.

Question 3. *Lastly, but most importantly, according to the result of NAP-XPS, the graphene started to fail to protect CoNi nanoparticles from its oxidation at 150 °C. To me, this shows that graphene is not fully attached to CoNi. Many previous reports showed that graphene on metal substrates can stay up to several hundred degrees of Celsius. The failure of protecting CoNi nanoparticles at this low temperature possibly indicates that there can be many defects on graphene layers. In fact, this graphene defects can easily trigger the activation of O₂ adsorption/dissociation. The role of graphene defects as active sites have been repeatedly reported and found to be very important for the study of graphene. I assume that most of the findings in this manuscript can be originated from the defect of graphene layers, instead of Pt-graphene interface. If the authors can come up with the solid evidence of defect free graphene layer in this study, this manuscript can be reviewed again. At the moment, I cannot recommend the publication of this manuscript.*

Author Reply: Thank you for your concern about the possible defects of graphene in our catalyst. We would like to answer it from the following three aspects.

Firstly, the graphenes of CoNi@NC and Pt/CoNi@NC can be proven to be intact by the acid leaching experiments, based on the fact that only one carbon vacancy can create a hole with a diameter of 2.1 Å which can allow Co²⁺ or Ni²⁺ (whose diameters are around 1.4 Å) be leached out. It should be noted that, a long-term acid leaching process had been carried out to obtain the CoNi@NC and CoNi@C samples according to our previous works (Angew. Chem. Int. Ed. 2015, 54, 2100-2104; Energy Environ. Sci. 2016, 9, 123-129; Energy Environ. Sci. 2020, 13, 119-126), during which the not fully encapsulated metals can be removed, thereby leave some hollow graphene cages as seen from the TEM images (Fig. 1a, Supplementary Fig. 1, and Supplementary Fig. 2a-2b). Here for your question, we did a further acid leaching process on the obtained CoNi@NC and Pt/CoNi@NC samples in a 10 wt% HCl aqueous solution at room temperature for 12 h, and the Co and Ni residuals determined by ICP-OES in the filtrate were negligible for both. It means that the graphene layer is intact to protect metals from acid leaching, and this is why CoNi@NC can also be used as a HER electrocatalyst in H₂SO₄ electrolyte (Angew. Chem. Int. Ed. 2015, 54, 2100-2104).

Secondly, the relatively low oxidation temperature (150 °C) of graphene in our catalyst is due to the unique electron penetration effect of this kind of catalyst (Angew. Chem. Int. Ed. 2020, 59, 15294-15297). The encapsulated metals can donate electrons to the outer graphene and then

decrease the local work function of graphene as we originally found in our previous work (Angew. Chem. Int. Ed. 2013, 52, 371-375), thus activating the inert graphene for chemical reactions. It should also be noted that this 150 °C is the onset temperature for oxidation, which is rational compared with those reported in literatures. For example, for a flat graphene on an inert SiO₂/Si substrate, O₂ oxidation at 200 to 300 °C can create strong hole doping in graphene (Nano Lett. 2008, 8, 1965-1970). Furthermore, the oxidation of graphene is highly sensitive to substrates (J. Phys. Chem. Lett. 2016, 7, 867-873; Nanoscale 2016, 8, 11494-11502). Zhang et al. found that graphene remains inert on SiO₂ and *h*-BN but becomes increasingly weak against oxidation on metal substrates because of enhanced charge transfer and chemical interaction between them (J. Phys. Chem. Lett. 2016, 7, 867-873), similar to the electron penetration effect in our work. They experimentally confirmed that graphene can be easily oxidized on fully covered Cu foil at 270 °C (not the onset temperature) in air, and their DFT calculations implied that Co and Ni foils can promote the oxidation of graphene to a larger extent comparing with Cu foil. CoNi-alloy NPs used in our case should be more active than the single-metal foils. Thus, we can say that the 150 °C for the oxidation of the graphene over 5-nm CoNi NPs is rational and reasonable. Actually, the intact graphene has already protected the CoNi NPs from oxidation below 150 °C, because our NAP-XPS results indicate that the small CoNi NPs will be easily oxidized at near room temperature without the protection of graphene (Supplementary Fig. 10).

Thirdly, we would like to discuss the role of graphene defect in CO oxidation. Indeed, graphene defect can activate O₂ as we also found before (Chem. Commun. 2011, 47, 10016-10018). Thus, we prepared Pt NPs on defect-rich graphene from graphene oxide according to our previous work (J. Catal. 2019, 377, 524-533), and then performed CO oxidation over this Pt/rGO catalyst. As shown in Fig. R22, we can see that its activity is between those over Pt/CNT and Pt/CB, indicating that the defect alone cannot effectively activate O₂ at room temperature and the underneath CoNi NPs are still the key to the high room-temperature activity. If one assumes that the role of graphene defect is to expose CoNi for directly activating O₂, actually we have confirmed that CoNi NPs will be oxidized at room temperature once exposed to O₂ and then lose the effectiveness to promote the activity of CO oxidation (Fig. 2d). In conclusion, as we stated on Page 8 of the manuscript, the intact graphene layer is very important for the keep of CoNi in the metallic state and thus for the effectiveness of the electron penetration effect on the room-temperature activity of CO oxidation.

Fig. R22 Temperature-dependence CO conversion in CO oxidation reaction over the pre-reduced catalysts. 1% CO and 20% O₂ in He (1 bar). Space velocity: 60000 mL·g⁻¹·h⁻¹.

REVIEWER COMMENTS

Reviewer #1 (Remarks to the Author):

I appreciate the thorough responses and additional data, tables, and figures from the authors. I have no further comments or reservations prior to publication of the revised manuscript.

Reviewer #2 (Remarks to the Author):

Overall, the authors have addressed most of the comments quite thoroughly with additional DFT calculations on enlarged nano-strip models and different CO coverage effects. The nano-strip models have shown consistency in results with the smaller original graphene cage models. The CO coverage effect on interfacial O₂ adsorption has shown to weaken O₂ binding energies within a reasonable range of <0.2 eV.

However, there is still some clarification that could be made regarding the authors' response to Question 3. For the suggested reversed-CoNi@C model, the interfacial O₂ adsorption strengthened by a significant amount (0.38 eV), while the Bader charge of Pt₄ is not much different (only 0.02). Therefore, it seems that other factors may be affecting the interfacial O₂ adsorption other than solely the electron penetration to Pt₄ (otherwise the Bader charge should have a meaningful difference in value to account for the 0.38 eV difference in binding energy). As the interfacial O₂ adsorption involves chemisorption of O atoms at both Pt and C, perhaps the electronic structure properties of the C atoms in proximity of the binding site (or directly binding to an O atom) or looking into O₂ binding on CoNi@C without Pt clusters may help to more fully understand why interfacial O₂ adsorption occurs on the system studied.

Reviewer #3 (Remarks to the Author):

The authors answered my previous questions with detailed explanation. So, I don't have any further questions in regard to those. However, I found the other question in the response letter. In the answer to question 6 of reviewer #1, the author showed the AP-XPS spectra of Pt 4f 7/2 in Fig. R10, claiming no significant change in Pt element. I strongly disagree with this argument. In Fig. R10, the peak of Pt 4f moves to the lower binding direction when the temperature reaches to 100C. The estimated peak position of Pt 4f is ~70.8 eV, which corresponds to the metallic states of bulk Pt 4f. (Please, find the figure attached.) After 100C, the spectra shift back to higher binding direction, possibly indicating the oxidation of Pt atom under oxygen rich CO oxidation condition. So, now it makes me think about the validity of XPS analysis and its interpretation.

First of all, from the first look, the line shape of Pt at T=25C shows the presence of oxidation. It is well known that oxidized Pt shows the increased intensity at higher binding energy side. This means the reactivity of Pt/CoNi@NC measured at 25C is mostly coming from the Pt-oxide, not from the metallic states of Pt mediated by CoNi. I wonder if the authors can answer to my question on this. Of course, cautions need to be made when you interpret XPS spectra of Pt nanoparticles. Many other effects can exist to influence the line shape of XPS, such as final state effect, charge transfer.

Out of my own curiosity, my own guess to this question is the reduction of charge transfer from CoNi to Pt at 100C. As temperature increases, the interaction between Pt and CoNi is being reduced and the amount of charge transfer is no longer there, meaning reduced charge donation through graphene

layer. Then, with the picture of rigid band model, the d-band of Pt moves closer to Fermi level, resulting in lower binding energy shift of Pt spectra. Of course, in this case, the confirmation should be made in regard to the assignment of Pt 4f peak at 71.3 eV as metallic state, which can be possibly due to the intermetallic bonding of Pt and CoNi via graphene. However, this requires a strong interaction between Pt and CoNi, similar to that of alloys.

Response to the reviewers' comments

We are grateful for the reviewers' helpful comments and suggestions. The revised part is highlighted in yellow in the file of "Revised Supplementary Information".

Reviewer #1

Comments: I appreciate the thorough responses and additional data, tables, and figures from the authors. I have no further comments or reservations prior to publication of the revised manuscript.

Author Reply: Thank you again for your valuable suggestions and the recommendation for publication.

Reviewer #2

Comments: Overall, the authors have addressed most of the comments quite thoroughly with additional DFT calculations on enlarged nano-strip models and different CO coverage effects. The nano-strip models have shown consistency in results with the smaller original graphene cage models. The CO coverage effect on interfacial O₂ adsorption has shown to weaken O₂ binding energies within a reasonable range of <0.2 eV.

However, there is still some clarification that could be made regarding the authors' response to Question 3. For the suggested reversed-CoNi@C model, the interfacial O₂ adsorption strengthened by a significant amount (0.38 eV), while the Bader charge of Pt₄ is not much different (only 0.02). Therefore, it seems that other factors may be affecting the interfacial O₂ adsorption other than solely the electron penetration to Pt₄ (otherwise the Bader charge should have a meaningful difference in value to account for the 0.38 eV difference in binding energy). As the interfacial O₂ adsorption involves chemisorption of O atoms at both Pt and C, perhaps the electronic structure properties of the C atoms in proximity of the binding site (or directly binding to an O atom) or looking into O₂ binding on CoNi@C without Pt clusters may help to more fully understand why interfacial O₂ adsorption occurs on the system studied.

Author Reply: Thank you for your insightful comment. According to your suggestion, we have analyzed the electronic structures of the carbon sites for O₂ adsorption in both models to further understand the decreased adsorption energy of O₂ in the case of the CoNi-reversed model. Similar to the small difference in the charge of Pt, the Bader charge difference of the carbon sites

between the two models is also very small (-0.004 e⁻). The 2*p* density of states (DOS) of the carbon sites also have no obvious difference. Thus, the activation of the carbon surface and Pt-graphene interface is not much affected by the local structure of the encapsulated metal. However, the adsorption of O₂ at the interface may induce local structural changes in the vicinity of the adsorption site depending on the element type, component, and size of the encapsulated metal, as found in our previous work (*Angew. Chem. Int. Ed.* 2013, 52, 371; *Energy Environ. Sci.* 2016, 9, 123; *Nano Energy* 2018, 52, 494). This could lead to fluctuated adsorption energies at the interface though the electronic structures of the adsorption sites are similar in the two models.

Therefore, we further analyzed the O₂ adsorption at the interface by decoupling the adsorption energy into two parts, $\Delta E_{total} = \Delta E_{deform} + \Delta E_{bonding}$. ΔE_{deform} is the deformation energy of the catalyst structure after O₂ adsorption, which is calculated by $\Delta E_{deform} = E_{catalyst}^{deformed} - E_{catalyst}^{initial}$, where $E_{catalyst}^{initial}$ and $E_{catalyst}^{deformed}$ are the energies of the initial structure before O₂ adsorption and deformed structure after O₂ adsorption, respectively. $E_{catalyst}^{deformed}$ was calculated based on the adsorption structure with O₂ removed without further structural relaxation. The $\Delta E_{bonding}$ is the bonding energy between O₂ and the deformed structure, which is calculated by $\Delta E_{bonding} = E_{O_2/catalyst} - E_{catalyst}^{deformed}$. From the results listed in Table R1, we can see that the enhanced adsorption of O₂ for the CoNi-reversed model (-0.38 eV) mainly comes from the difference in the ΔE_{deform} (-0.24 eV) rather than that in the $\Delta E_{bonding}$ (-0.14 eV). Upon O₂ adsorption on the carbon sites, the hybridization type of the carbon atoms turns from sp² to sp³, which induce different structural changes in both the Pt cluster and encapsulated metal depending on the configuration of CoNi, thus leading to different ΔE_{deform} .

Table R1. The adsorption energies of O₂ and the contributions from deformation (ΔE_{deform}) and bonding ($\Delta E_{bonding}$) on Pt/CoNi@C and Pt/reversed-CoNi@C.

Energy (eV)	Pt/CoNi@C	Pt/reversed-CoNi@C	Difference
ΔE_{deform}	1.32	1.08	-0.24
$\Delta E_{bonding}$	-1.69	-1.83	-0.14
ΔE_{total}	-0.37	-0.75	-0.38

According to your suggestion, we calculated the adsorption of O₂ on CoNi@NC without the presence of Pt₄. For a better comparison, we tried the atop-mode adsorption of O₂ on one carbon atom and found that the adsorption is unstable, probably because two binding sites are usually needed to saturate the biradical O₂. Thus, Pt is essential to construct the active interface for effective adsorption and activation of O₂ in this system.

Reviewer #3

Comments: The authors answered my previous questions with detailed explanation. So, I don't have any further questions in regard to those. However, I found the other question in the response letter. In the answer to question 6 of reviewer #1, the author showed the AP-XPS spectra of Pt 4f 7/2 in Fig. R10, claiming no significant change in Pt element. I strongly disagree with this argument. In Fig. R10, the peak of Pt 4f moves to the lower binding direction when the temperature reaches to 100C. The estimated peak position of Pt 4f is ~70.8 eV, which corresponds to the metallic states of bulk Pt 4f. (Please, find the figure attached.) After 100C, the spectra shift back to higher binding direction, possibly indicating the oxidation of Pt atom under oxygen rich CO oxidation condition. So, now it makes me think about the validity of XPS analysis and its interpretation.

First of all, from the first look, the line shape of Pt at T=25C shows the presence of oxidation. It is well known that oxidized Pt shows the increased intensity at higher binding energy side. This means the reactivity of Pt/CoNi@NC measured at 25C is mostly coming from the Pt-oxide, not from the metallic states of Pt mediated by CoNi. I wonder if the authors can answer to my question on this. Of course, cautions need to be made when you interpret XPS spectra of Pt nanoparticles. Many other effects can exist to influence the line shape of XPS, such as final state effect, charge transfer.

Out of my own curiosity, my own guess to this question is the reduction of charge transfer from CoNi to Pt at 100C. As temperature increases, the interaction between Pt and CoNi is being reduced and the amount of charge transfer is no longer there, meaning reduced charge donation through graphene layer. Then, with the picture of rigid band model, the d-band of Pt moves closer to Fermi level, resulting in lower binding energy shift of Pt spectra. Of course, in this case, the confirmation should be made in regard to the assignment of Pt 4f peak at 71.3 eV as metallic

state, which can be possibly due to the intermetallic bonding of Pt and CoNi via graphene. However, this requires a strong interaction between Pt and CoNi, similar to that of alloys.

Author Reply: Thank you for your comment on the NAP-XPS spectra of Pt 4f_{7/2} (Figure S9). Firstly, we would like to point out that the peaks displayed in the previous Figure S9 are too broad to observe the peak position accurately. Here we present a screenshot of the normalized display of these original spectra in CasaXPS software (Figure R1), from which we can see that the low-energy edges of them are overlapped and their peak positions fall into the range of 71.3 ± 0.1 eV, which is within the allowed error range of XPS test. Thus, it is reasonable to describe that there is no obvious change for the signal of Pt 4f_{7/2} during the in-situ reaction from 25 to 210 °C (Page 6 of the manuscript). For a clear observation, we replaced the previous chart in Figure S9 with the current Figure R1 (Page 10 of SI).

Figure R1. Normalized display of Pt 4f_{7/2} XPS spectra in CasaXPS software. During the NAP-XPS testings, the in-situ reaction was performed over Pt/CoNi@NC in a flow of 0.067 mbar CO and 1.13 mbar O₂ at different reaction temperatures (25-210 °C), following with the reduction process.

Furthermore, we can confirm that the Pt 4f_{7/5} peak at 71.3 eV in our work does correspond to the Pt in metallic state. Before our test, the instrument was calibrated with Ar-sputtered polycrystalline Ag by using monochromatic Al K α radiation (1486.6 eV, 20 W source power). The Ag 3d_{5/2} peak was calibrated to 368.0 eV, and the FWHM is 0.49 eV at 24 kcps signal above background. After that, a Pt (111) single crystal was tested in a separate XPS experiment, and the position of Pt 4f_{7/5} peak is at 71.16 eV. The 0.14-eV higher binding energy observed for our Pt NPs (2 nm) should be due to the final state effect (*Phys. Rev. B* 1983, 27, 748; *Surf. Sci.* 1984, 140, 151) as you mentioned. This slight shift is not an indication of oxidized Pt, because after H₂ reduction the peak position is still at 71.3 eV (Figure S8 and Figure S9), while Pt²⁺ usually has a 1-eV higher binding energy than Pt⁰ (*Anal. Chem.* 1975, 47, 586). It is also not due to the formation of PtCo or PtNi alloys, which is not the case for our graphene-isolated catalyst. Even for a pure Pt film, the Pt 4f_{7/2} peak can also appear at 71.3 eV after calibration with Au 4f_{7/2} to 84.0 eV (*Anal. Chem.* 1975, 47, 586). Thus, it is reasonable to attribute the Pt 4f_{7/2} peak at 71.3 eV to metallic Pt.

REVIEWER COMMENTS

Reviewer #2 (Remarks to the Author):

The authors have addressed all my previous comments. Their explanation that the local structural changes rather than different electronic structures are the reason for different O₂ binding energies of the CoNi@C and reversed-CoNi@C is consistent with additional DFT calculations performed. I have no further comments.

Reviewer #3 (Remarks to the Author):

First of all, I am not questioning about the calibration procedure. The calibration procedure with Ag is a very well-known standard procedure and I do believe that authors follow the standard procedure properly. Nonetheless, whenever the XPS spectra are measured, Fermi level is also checked as another internal reference. In metal, the Fermi level is the position of zero binding energy. I wonder if the authors have measured the Fermi level also.

Second, I don't agree with author's suggestion of using Figure R1 copied from CasaXPS software screen. In the Figure R1, authors made the reference point on binding energy of 73 eV and compared the highest peak positions of Pt 4f. However, this is not how XPS spectra should be compared. Any experienced XPS users will be able to tell that this procedure is not correct. The background are not matching in both high and low binding energy side. Of course, I understand the difficulty of matching background of XPS spectra obtained from nanoparticle system due to many complex XPS effect. For this reason, I prefer the previous figure that authors used for first revision. In previous plot, I assumed the authors applied the simple normalization on raw data. You can see more clear evolution of Pt 4f peak as a function of temperature in previous plot. (See the attachment.)

Third, when I mentioned about the possible oxidation state of Pt 4f, I did NOT mean that the presence of PtO or PtO₂ oxide. Pt is a noble metal, which does not form stable oxide at RT temperature. What I meant from previous comments was the presence of oxide peak, PtO_x, under room temperature. If the sample is exposed to air for any brief time, the contamination of water from atmosphere can generate metastable Pt oxide. Considering the high surface areas of nanoparticles, the formation of surface oxide is certainly possible. So, I suspected the high background of Pt 4f as a sign of PtO_x. Of course, as reviewer pointed out, the final state effect can interfere with this and generate high background. I will get back to this discussion later.

Fourth, I would like to discuss about the binding energy position of Pt 4f. The author claimed that the metallic state of Pt 4f in Pt(111) as 71.16 eV. In fact, it is not easy to compare the peak position of Pt 4f from Pt(111) single crystal to that of Pt nanoparticle. In single crystal surface, if cleaned properly, the peak from the surface state exist at ~70.6 eV. (Phys. Chem. Chem. Phys., 2014, 16, 23564–23567) My estimate is that authors do not see the surface state due to low cross section of photon energy from Al source. If both peaks from bulk and surface states exist together, one could find the peak position of Pt 4f at 71.16 eV, as authors claimed. So, I can accept the claim of authors on this aspect.

To summary, I agree most of the authors' claims in response letter, except the pure metallic states of Pt based on XPS analysis in Figure R1. Author claimed that there is no alloy formation between Pt and CoNi alloys. However, the peak position at 71.3 eV at RT and 25 C and 50 C can be interpreted as either interfacial oxide or alloy formation. Several previous results, e.g. RSC Adv., 2016, 6, 71501–71506, NATURE COMMUNICATIONS | (2019)10:1743, Pt 4f peak moves to the higher binding direction

as Pt forms nanoparticle alloys. Another possibility is that Pt nanoparticles are heavily influenced by interfacial graphene and also CoNi, resulting in broad metallic states, without alloy formation. I would like to see the solid proof of metallic state of Pt, other than XPS.

Fig. Revision 1.

Fig. Revision 2

Response to the reviewers' comments

Reviewer #2

Comments: The authors have addressed all my previous comments. Their explanation that the local structural changes rather than different electronic structures are the reason for different O₂ binding energies of the CoNi@C and reversed-CoNi@C is consistent with additional DFT calculations performed. I have no further comments.

Author Reply: Thank you again for your valuable suggestions.

Reviewer #3

Comments: First of all, I am not questioning about the calibration procedure. The calibration procedure with Ag is a very well-known standard procedure and I do believe that authors follow the standard procedure properly. Nonetheless, whenever the XPS spectra are measured, Fermi level is also checked as an another internal reference. In metal, the Fermi level is the position of zero binding energy. I wonder if the authors have measured the Fermi level also.

Author Reply: Thank you for your helpful comments. We didn't finely scan the valance band spectra during the in-situ NAP-XPS testings over our catalyst, but we measured the survey spectra from 0 to 1000 eV, from which we can roughly see that there is almost no significant change for the Fermi level at 0 eV during the testings (Figure R1).

Figure R1. Survey spectra from 0 to 20 eV during in-situ NAP-XPS testings over Pt/CoNi@NC

Second, I don't agree with author's suggestion of using Figure R1 copied from CasaXPS software screen. In the Figure R1, authors made the reference point on binding energy of 73 eV and compared the highest peak positions of Pt 4f. However, this is not how XPS spectra should be compared. Any experienced XPS users will be able to tell that this procedure is not correct. The background are not matching in both high and low binding energy side. Of course, I understand the difficulty of matching background of XPS spectra obtained from nanoparticle system due to many complex XPS effect. For this reason, I prefer the previous figure that authors used for first revision. In previous plot, I assumed the authors applied the simple normalization on raw data. You can see more clear evolution of Pt 4f peak as a function of temperature in previous plot. (See the attachment.)

Author Reply: Thank you for the detailed explanation. We adopted the previous figure as Figure S9 in SI according to your suggestion.

Third, when I mentioned about the possible oxidation state of Pt 4f, I did NOT mean that the presence of PtO or PtO₂ oxide. Pt is a noble metal, which does not form stable oxide at RT temperature. What I meant from previous comments was the presence of oxide peak, PtO_x, under room temperature. If the sample is exposed to air for any brief time, the contamination of water from atmosphere can generate metastable Pt oxide. Considering the high surface areas of nanoparticles, the formation of surface oxide is certainly possible. So, I suspected the high background of Pt 4f as a sign of PtO_x. Of course, as reviewer pointed out, the final state effect can interfere with this and generate high background. I will get back to this discussion later.

Author Reply: We totally agree with your assumption that the surface atoms of Pt NPs can be partially oxidized to PtO_x when exposing to air. During our in-situ XPS testings, there can also in-situ form some metastable PtO_x surface species in the presence of O₂ and CO. However, after the in-situ H₂ reduction (without exposing to air or reactant gases), the peak position of Pt 4f_{7/2} is still at around 71.3 eV (Figure S8 and Figure S9), indicating that the amount of the possibly formed PtO_x species may not be sufficient enough to cause a significant shift of the binding energy of Pt 4f, and the Pt NPs should be mainly in the metallic state, at least for the bulk phase.

Fourth, I would like to discuss about the binding energy position of Pt 4f. The author claimed that the metallic state of Pt 4f in Pt(111) as 71.16 eV. In fact, it is not easy to compare the peak position of Pt 4f from Pt(111) single crystal to that of Pt nanoparticle. In single crystal surface, if cleaned properly, the peak from the surface state exist at ~70.6 eV. (Phys. Chem. Chem. Phys., 2014, 16, 23564–23567) My estimate is that authors do not see the surface state due to low cross section of photon energy from Al source. If both peaks from bulk and surface states exist together, one could find the peak position of Pt 4f at 71.16 eV, as authors claimed. So, I can accept the claim of authors on this aspect.

Author Reply: Thank you for your professional interpretation and understanding.

To summary, I agree most of the authors' claims in response letter, except the pure metallic states of Pt based on XPS analysis in Figure R1. Author claimed that there is no alloy formation between Pt and CoNi alloys. However, the peak position at 71.3 eV at RT and 25 C and 50 C can be interpreted as either interfacial oxide or alloy formation. Several previous results, e.g. RSC Adv., 2016, 6, 71501–71506, NATURE COMMUNICATIONS | (2019)10:1743, Pt 4f peak moves to the higher binding direction as Pt forms nanoparticle alloys. Another possibility is that Pt nanoparticles are heavily influenced by interfacial graphene and also CoNi, resulting in broad metallic states, without alloy formation. I would like to see the solid proof of metallic state of Pt, other than XPS.

Author Reply: Thank you for sharing these literatures with us. We noticed that 71.32 eV was found for pure Pt NPs (RSC Adv. 2016, 6, 71501–71506) and 71.2 eV was found for Pt in commercial Pt/C catalyst (Nat. Commun. 2019, 10, 1743). The latter paper also stated that the 71.2 eV for Pt/C is an indicative of Pt⁰, but their focused catalyst is not PtCo alloy but single-atom Pt on Co_{0.85}Se alloy. For the PtNi alloy in the former paper, the Pt 4f peak moves to the lower binding direction (negative shift) when forming alloy. It seems that the shift direction of Pt 4f core level is metal-dependent, because Pt presented a positive shift in PtCo alloy as you suggested but had almost no shift in PtCu alloy (Surf. Interface Anal. 2000, 30, 475–478). Nevertheless, PtCo or PtNi alloy will not be formed in our case, because Pt NPs (Chem. Mater. 2000, 12, 1622–1627) and CoNi@NC were separately prepared by different methods and then mixed together in ethanol at room temperature to obtain the Pt/CoNi@NC catalyst (details see Methods in manuscript). Thus, according to the two literatures you shared and more others (Nat.

Catal. 2021, 4, 312–321; *Nat. Commun.* 2021, 12, 3021; *Nat. Catal.* 2020, 3, 376–385; *Nat. Commun.* 2016, 7, 10922; *Nat. Mater.* 2016, 15, 564–569; *Science* 2003, 301, 935–938) and based on our in-situ XPS data (Figure S8 and Figure S9), it is reasonable to attribute the Pt 4f_{7/2} peak at 71.3 eV to metallic Pt for our small Pt NPs. Besides the final state effect, it may also be partially caused by the strong interaction between Pt NPs and graphene with CoNi NPs inside, as you proposed. Other than XPS, our HAADF-STEM (Figure 1d) and XAS (Figure 1f) data can also indicate that the Pt NPs should be mainly in the metallic state, at least for the bulk phase.

REVIEWERS' COMMENTS

Reviewer #3 (Remarks to the Author):

The authors have addressed all my previous comments and recommend the publication.

Response to the reviewers' comments

Reviewer #3

Comments: The authors have addressed all my previous comments and recommend the publication.

Author Reply: Thank you again for your valuable suggestions.